# Genetic analyses implicate complex links between adult testosterone levels and health and disease

Jaakko T. Leinonen [1], Nina Mars [1,26], Leevi E. Lehtonen[1,26], Ari Ahola-Olli[1,26], Sanni Ruotsalainen [1], Terho Lehtimäki [2,3,4], Mika Kähönen[3,5], Olli Raitakari[6,7,8], FinnGen Consortium*, Terhi Piltonen [9], Mark Daly[1,10], Tiinamaija Tuomi[1,11,12,13,14], Samuli Ripatti [1,10,15], Matti Pirinen [1,15,16] & Taru Tukiainen [1✉]

## Abstract

**Background** Testosterone levels are linked with diverse characteristics of human health, yet, whether these associations reflect correlation or causation remains debated. Here, we provide a broad perspective on the role of genetically determined testosterone on complex diseases in both sexes.

**Methods** Leveraging genetic and health registry data from the UK Biobank and FinnGen (total $N = 625,650$), we constructed polygenic scores (PGS) for total testosterone, sex-hormone binding globulin (SHBG) and free testosterone, associating these with 36 endpoints across different disease categories in the FinnGen. These analyses were combined with Mendelian Randomization (MR) and cross-sex PGS analyses to address causality.

**Results** We show testosterone and SHBG levels are intricately tied to metabolic health, but report lack of causality behind most associations, including type 2 diabetes (T2D). Across other disease domains, including 13 behavioral and neurological diseases, we similarly find little evidence for a substantial contribution from normal variation in testosterone levels. We nonetheless find genetically predicted testosterone affects many sex-specific traits, with a pronounced impact on female reproductive health, including causal contribution to PCOS-related traits like hirsutism and post-menopausal bleeding (PMB). We also illustrate how testosterone levels associate with antagonistic effects on stroke risk and reproductive endpoints between the sexes.

**Conclusions** Overall, these findings provide insight into how genetically determined testosterone correlates with several health parameters in both sexes. Yet the lack of evidence for a causal contribution to most traits beyond sex-specific health underscores the complexity of the mechanisms linking testosterone levels to disease risk and sex differences.

## Plain language summary

Hormones, such as testosterone, travel around the body communicating between the different parts. Testosterone is present at higher levels in men, but also present in women. Variable testosterone levels explain some differences in human traits and disease prevalence. Here, we study how adult testosterone levels relate to health and disease. Genetic, i.e. inherited, differences in testosterone levels contribute to many traits specific to men or women, such as women's reproductive health, hormonal cancers, and hair growth typical in males. However, testosterone levels do not appear as a major cause of most traits studied, including psychiatric diseases and metabolic health. Normal variation in baseline testosterone levels thus seems to have a relatively minor impact on health and disease.

[1] Institute for Molecular Medicine Finland (FIMM), HiLIFE, University of Helsinki, Helsinki, Finland. [2] Department of Clinical Chemistry, Finnish Cardiovascular Research Center, Tampere, Finland. [3] Faculty of Medicine and Health Technology, Tampere University, Tampere, Finland. [4] Fimlab Laboratories, Tampere, Finland. [5] Department of Clinical Physiology, Finnish Cardiovascular Research Center, Tampere, Finland. [6] Centre for Population Health Research, University of Turku and Turku University Hospital, Turku, Finland. [7] Research Centre of Applied and Preventive Cardiovascular Medicine, University of Turku, Turku, Finland. [8] Department of Clinical Physiology and Nuclear Medicine, Turku University Hospital, Turku, Finland. [9] Department of Obstetrics and Gynecology, PEDEGO Research Unit, Medical Research Centre, Oulu University Hospital, University of Oulu, Oulu, Finland. [10] Broad Institute of MIT and Harvard, Cambridge, MA, USA. [11] Department of Endocrinology, Abdominal Centre, Helsinki University Hospital, Helsinki, Finland. [12] Folkhalsan Research Center, Helsinki, Finland. [13] Research Programs Unit, Clinical and Molecular Metabolism, University of Helsinki, Helsinki, Finland. [14] Lund University Diabetes Centre, Department of Clinical Sciences Malmö, Lund University, Malmö, Sweden. [15] Department of Public Health, Faculty of Medicine, University of Helsinki, Helsinki, Finland. [16] Helsinki Institute for Information Technology HIIT and Department of Mathematics and Statistics, University of Helsinki, Helsinki, Finland. [26] These authors contributed equally: Nina Mars, Leevi E. Lehtonen, Ari Ahola-Olli. *A list of authors and their affiliations appears at the end of the paper. ✉email: taru.tukiainen@helsinki.fi

Testosterone (T) is the male sex hormone responsible for regulation of development of primary and secondary male sexual characteristics. Individual variation in T levels has been suggested to shape human physiology broadly, including effects on disease risk in both men and women[1–3]. Epidemiological studies and randomized clinical trials for T replacement therapy have observed associations between serum T levels and various traits ranging from type 2 diabetes (T2D) and cardiovascular disease to body composition and behavior[1–11]. Yet, these studies have yielded partly mixed results, and, in many instances, the proposed relationships between T, complex traits and disease remain elusive[2,7–11].

Besides the disease links, T is a known driver for sex differences. After puberty, men and women differ extensively with respect to their average T levels, with males showing roughly 7–15-fold higher serum total T concentrations[1,12]. This difference largely results from the testicular T production in males that far exceeds the amount of T produced in the ovaries and the adrenal gland in females, and is known to directly contribute to variation in, for instance, body composition between the sexes[1,12].

In the human body, the majority of T is bound to a carrier molecule, whereas only a small fraction (1–3%) of this total T exists as free T[13–16]. Free T is considered to represent the most potent form of T in terms of biological activity, and although extensively debated may have different clinical significance than total T[13–16]. Most of the remaining T in circulation is tightly bound by sex hormone-binding globulin (SHBG), and the bulk of the rest remains attached to carrier proteins like serum albumin (Fig. 1a)[4,13,14,16].

Complex physiological processes regulate circulating T levels, allowing T levels to fluctuate based on internal and external stimuli on a daily basis[17,18]. However, twin studies indicate that heritability of serum T is relatively high in both sexes, being up to 65% in males[19,20]. Until recently, few genetic variants were shown to associate with circulating T levels and related traits in the general population[4,19,21,22]. Yet, through the emergence of genetic and biomarker data from large biobanks, currently more than a hundred, mostly autosomal, loci for serum total T, SHBG and free T have been identified, with evidence for sex-specific genetic effects[23–25].

Recent efforts have used genetics to address the potential causal contribution of adult T to selected complex traits in both sexes, including T2D, body composition and hormonal cancers, or have examined the effects of free T more broadly in males based on the UK Biobank data[23,24,26–28]. Here, we extend these investigations combining data from the UK Biobank ($N = 408,186$) and FinnGen ($N = 217,464$). Leveraging the extensive healthcare registry data of FinnGen, sex-specific polygenic scores (PGS), and public data from genome-wide association studies (GWAS), we studied the role of T levels in both sexes across traits ranging from metabolic conditions and sex-specific reproductive disorders to neurological and behavioral endpoints.

We illustrate that normal variation in genetically set T levels has only subtle effects to most human traits. For example, the association between T and many metabolic phenotypes may instead largely stem from shared etiology, challenging some previous findings. Yet we stress the unique genetic architecture of serum T levels, potentially reflecting sexually antagonistic effects of T on fitness, and emphasize the causal role of the hormone also for women's health.

## Methods

**Genotype and phenotype data from the UK Biobank.** The genetic association analysis was based on data from the UK

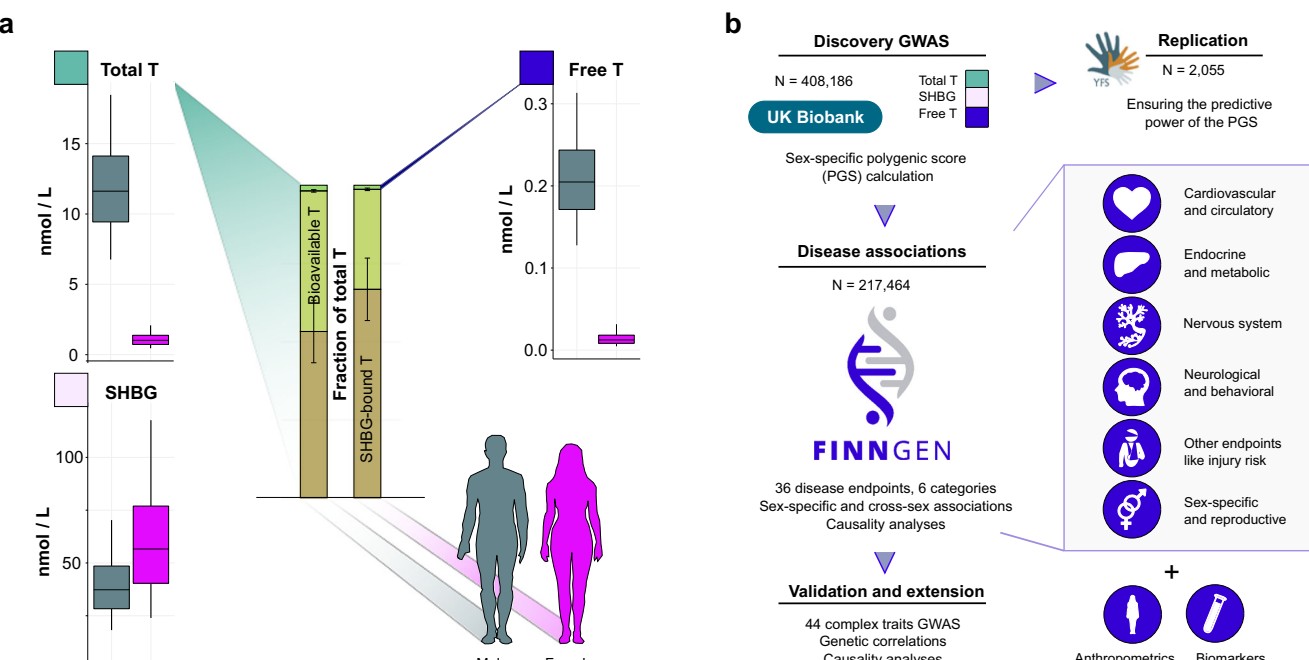

**Fig. 1 Illustration of the studied traits and study overview. a** The traits subjected to GWAS (total T, SHBG and free T) and distribution of these in the UK Biobank. The barplots illustrate division of serum total T to SHBG-bound (brown), and bioavailable fractions (green) including free T (bright green) for men (dark green) and women (pink). The three boxplots in **a** show sex-specific distributions for serum total T, SHBG and calculated free T levels for the UK Biobank participants included in the GWAS. These boxplots show median for each trait (black line), lower and upper quartiles (colored area of the box) and the error bars indicate 5 and 95% quantiles. **b** Overview of the study design to assess the contribution of T to health and disease using genetic approaches and biobank data. We conducted the discovery GWAS in the UK Biobank ($N = 408,186$), built sex-specific PGSs for the three T-related traits and validated the predictive ability of the PGSs in the Young Finns Study (YFS) ($N = 2055$). We then performed complex disease and trait associations using the PGS in FinnGen (release 5, $N = 217,464$), complementing these analyses with Mendelian Randomization analyses and extending these analyses with publicly available GWAS data from other cohorts.

Biobank, a population-based biobank consisting of 502,637 subjects (aged 37–73 years)[29]. Approximately 74% of the female participants in the cohort have reportedly undergone menopause. At recruitment, participants provided electronic signed consent. Ethics approval for the UK Biobank study was obtained from the North West Centre for Research Ethics Committee (11/NW/0382). All experiments were performed in accordance to relevant guidelines and regulations including the Declaration of Helsinki ethical principles for medical research. This study was run under UK Biobank application number 22627.

Multiple biochemical assays have been performed on the entire UK Biobank cohort, including measurements of serum testosterone, SHBG and serum albumin. These measurements were performed once for each participant for using Beckman Coulter DXI 800 Chemiluminescent Immunoassay, with competitive binding for testosterone and two-step sandwich for SHBG, referenced in https://biobank.ndph.ox.ac.uk/showcase/showcase/docs/serum_biochemistry.pdf. The document contains also details about the extensive QC procedures for all biochemical measurements in the UK Biobank. Analytical range for the immunoassays of total testosterone and SHBG were 0.35 to 55.52 and 0.33 to (226–242) nmol/L, respectively. Assay consistency was monitored by using internal QC samples between batches, and external quality assurance schemes against the ISO 17025:2005 standard. Notably, the reported testosterone and free T levels in the UK biobank are on average lower than for many other cohorts, for example YFS (Supplementary Data 2 and 6)[23,26], which may be partly accountable for UK Biobank measures being based on immunoassays instead of mass spectrometry.

We restricted our study to encompass 408,186 individuals from the white British subset. We removed outliers for genotype heterozygosity and missingness, as well as samples with sex chromosome aneuploidies, mismatches between reported and inferred sex, and samples that UK Biobank did not use in relatedness calculations[29]. We included related samples since our analysis method (BOLT-LMM) allows for their inclusion. To ensure that the UK Biobank measurements capture baseline heritable variation in adult T levels, we assessed correlation of testosterone measurements between two timepoints available for a subset of a UK Biobank population (7097 men and 5285 women included in the primary analyses), with the original samples collected between 2006 and 2010, and additional samples taken in 2012–2013. T levels from these two time points showed a high correlation ($R = 0.678$ for males and $R = 0.709$ for females, Supplementary Fig. 3).

For the GWASs, we used biochemically measured testosterone and SHBG, and calculated FAI and free T. For clarity, given the close relationship between free T and FAI, in the results section we concentrate on describing the results for free T, whereas the results for FAI are available in the supplementary data. For calculation of FAI, we used the formula $100 * Testosterone/SHBG$ (nmol/ml). Calculated free T was derived using the Vermeulen equation using directly measured albumin values for each participant[30,31].

All four traits were separated by sex and log transformed. Subjects with residuals values of $+/-5$ SD from the mean were excluded from the analyses, serving as a further QC step to exclude outliers potentially reflecting medical conditions or drug use affecting androgen levels. Inverse normalized values of the remaining residuals were used as phenotype values for the GWAS analyses. After the QC steps, our study included altogether 177,499 men and 205,141 women.

**Genetic association analysis and definition of the lead SNPs.** The GWAS analyses were performed using BOLT-LMM (v2.3.2)[32,33]. Imputed SNPs were restricted to variants with MAF $\geq 0.1$ % and imputation quality $\geq 0.7$[29]. We included both autosomal and X-chromosomal variants into the analysis, excluding pseudoautosomal regions (PARs). X-chromosomal effects in males were based on (0,2) allele dosage coding. 1000 Genomes European data was used as reference LD scores for calibrating the BOLT-LMM statistic. First 10 Principal Components were used as quantitative covariates in the runs. A linear regression model was fit for each trait with BMI and age as covariates, as well as menopause status for females. Genetic correlation analyses, heritability estimates and number of loci found implied the results remained consistent with different covariate configurations (Supplementary Fig. 10)[28,29]. Including up to 127 covariates (based on ref. [34]), e.g., assay center, dilution factors, blood draw time, and socioeconomic status indicators, or excluding related individuals from the analysis all showed negligible effects on the genetic findings (Supplementary Fig. 10).

Independent lead SNPs were selected for each chromosome by recursively taking the SNP with the lowest $p$-value (until none below the $p$-value threshold 5e-08 were left) from the GWAS summary statistics and removing all SNPs 500 kb on each side of it from the next round. The chromosomal positions of these 1 mb windows were stored, and overlapping windows were merged into the final list of loci. The SNP with the lowest $p$ value in each of these windows was selected as the lead SNP.

**Statistics and reproducibility.** Details of the variety of statistical methods used are described in the methods subsections found below. Our study design comprised of several steps to ensure robustness of the findings, and the conclusions of the study are based on multiple converging statistical analyses and vast datasets. In short, our study included confirming the reproducibility of the genetic findings from the UK Biobank ($N = 408,186$) (Supplementary Data 1) using the Young Finns Study ($N = 2,055$) (Supplementary Data 6), running discovery PGS and MR analyses in FinnGen ($N = 217,464$) (Supplementary Data 7 and 8) and replicating and extending these findings using publicly available GWAS summary statistics with data from tens of thousands to over a million participants (Supplementary Data 11–13).

**Pathway, tissue enrichment and co-localization analyses.** Tissue and gene set enrichment analyses were carried out with SNP2GENE and GENE2FUNC implemented in FUMA using default settings[35]. For testing in which tissues the genes residing in the GWAS loci were preferably expressed, we used the full distribution of SNP p-values and the GTEx v6 30 general tissue types. For pathway analysis, to assess whether the genes in the GWAS loci are overrepresented in pre-defined gene sets via hypergeometric tests, we selected manually curated KEGG-pathways[36]. For co-localization analyses to assess whether the genetic loci showed evidence for shared genetic effects between males and females, and to estimate the maximum posterior probability (MAP) for the loci being shared, we used gwas-pw[37].

**Replication in the Young Finns Cohort and calculation of PGS.** The Cardiovascular Risk in Young Finns Study (YFS) is a longitudinal follow-up of 3,596 subjects at baseline. The baseline survey was conducted in 1980 and subsequent follow-ups involving the whole sample were held in 1983, 1986, 2001, 2007, 2011, and 2017. The YFS study has been approved by the ethical committee of the Hospital District of Southwest Finland, all study subjects gave an informed consent, and the study has been conducted according to the principles of the Declaration of Helsinki. For testosterone and SHBG, we used data on 2001 follow-up (Subjects aged 24–39 years). A venous blood sample was drawn from antecubital vein after 12-h overnight fast. Serum was aliquoted and stored at

−70 degrees Celsius until analysis. In males, total testosterone quantification was performed in 2009 with competitive radio-immunoassay (Spectria Testosterone kit, Orion Diagnostica, Espoo, Finland) and Bio-Rad Lyphocheck control serums 1, 2, and 3 were used in quality control. Before quantification, serum aliquots had been melted three times. Total testosterone was quantified first and aliquots were re-frozen before SHBG quantification. In females, total testosterone quantification was performed in 2011. SHBG quantification for males was done in 2009 and for females in 2011 with Spectria SHBG IRMA kit (Orion Diagnostica, Espoo, Finland). Free testosterone was estimated using Vermeulen's formula. As albumin concentration was not available, we used fixed albumin concentration of 43 g/l.

In YFS, genotyping was performed at the Wellcome Trust Sanger Institute (UK) using customized Illumina Human Map 670k bead array. The custom content on 670k array replaced some poor performing probes on Human610 and added more CNV content. As quality control, we excluded individuals and probes with over 5% of missingness (--geno and --mind filters in Plink). Variants deviating from Hardy-Weinberg equilibrium ($p < 1 \times 10^{-6}$) and minor allele frequency below 1% were excluded. Related samples were excluded ($n = 51$) with pi-hat cut-off of 0.2. Total of 2442 individuals and 546,674 variants passed the quality control measures. The mean call rate across all included markers after the quality control was 0.9984. Next, imputation was performed using population-specific Sequencing Initiative Suomi (SISu) as reference panel. We examined the association of the UK Biobank GWAS lead SNPs with the corresponding T trait in YFS. If the annotated lead SNP was not available in YFS, we used LDstore2 (v2.0b)[38] to calculate LD in a 100 kb window around the lead SNP in the UK Biobank imputed data and selected the closest SNP with $R^2 > 0.8$ with the lead SNP as a proxy.

To construct PGSs we applied the LDpred[39] method to the sex-specific GWAS summary statistics from the UK Biobank for total T, SHBG, FAI and free T, using 1000 Genomes Europeans as LD reference and the default LD radius to account for LD. We then used the weights from the LDpred infinitesimal model to construct genome-wide PGSs for each individual in the YFS with Plink 2.0 (11 Feb 2018). Only variants imputed with high confidence (imputation INFO > 0.8) were included in PGS calculation. Variants in chromosomes 1–22 and chrX were included. In males, allele dosage of 2 was used for X-chromosomal haploid variants. To evaluate the PGS prediction accuracy in the YFS, we calculated the $R^2$ for each trait using using linear regression with z-score normalized PGS as predictor, age and 10 PCs as covariates and z-score normalized T trait as outcome.

### Estimation of heritability and calculation of the genetic correlations by LDSC.
SNP-based heritability for the studied T traits and genetic correlations between these and with 44 additional phenotypes were estimated using linkage disequilibrium score regression (LDSC), including autosomal variants only[40]. The summary statistics for the 44 traits were downloaded directly from the source repositories and analyzed locally, for the original sources please see references in Supplementary Data 11. For the genetic correlation analyses in LDSC, pre-computed LD Scores from 1000 Genomes Europeans, excluding the HLA region were used. For 23 traits we performed the analyses using sex-specific GWAS results and compared these to data from sex-combined GWAS (Supplementary Data 13). Generally, genetic correlation results using either sex-specific or sex-combined GWAS data were highly similar.

### Disease associations in FinnGen.
To assess if the PGS for studied traits associate with disease risk we utilized the FinnGen study (data freeze 5), consisting of 217,464 (94,478 men, 122,986 women)[41]. FinnGen is comprised of Finnish prospective epidemiological and disease-based cohorts and voluntary biobank samples collected by hospital biobanks. The genotypes have been linked to national hospital discharge (available from 1968), death (1969–), cancer (1953–) and medication reimbursement (1964–) registries as well as the registry on medication purchases (1995–). Although specific menopause status for the participants is not available, roughly 67% of the females in R5 ($N = 84,521$) are estimated to have undergone menopause at the end of the follow-up (age >50). The samples were genotyped with Illumina and Affymetrix arrays (Illumina Inc., San Diego, and Thermo Fisher Scientific, Santa Clara, CA, USA). The genotypes have been imputed with using the SISu v3 population-specific reference panel developed from high-quality data for 3775 high-coverage (25–30×) whole-genome sequencing in Finns. The detailed genotype imputation workflow can be found at https://doi.org/10.17504/protocols.io.xbgfijw. The dataset uses genome build 38 (hg38). A full list of FinnGen consortium members can be found from Supplementary Information.

For PGS analyses, we used same variant weights (LDpred infinitesimal model) as for YFS, and calculated genome-wide PGSs for each individual with PLINK2 (v2.00a2.3LM). Variants in chromosomes 1–22 and chromosome X (imputed with high confidence, imputation INFO ≥ 0.7) were included (total number of variants ranging from 6,535,263 for female total T to 6,536,405 for female SHBG) and we used genotype dosages to incorporate imputation uncertainty. In males, allele dosage of 2 was used for X-chromosomal haploid variants. We studied the PGS associations to 36 disease endpoints with potential links to androgens, representing six loosely defined disease categories. For details of the studied phenotypes see Supplementary Data 7, www.finngen.fi and risteys.finngen.fi. Cox proportional hazards models were used for estimating hazard ratios (HRs) and 95% CIs, with age as the time scale and 10 first principal components of ancestry and genotyping batch as covariates. The proportionality assumption for Cox models was assessed with Schoenfeld residuals and log-log plots. We performed power analyses on the Cox models using the power-EpiCont.default from powerSurvEpi package in R. We estimated that with the selected p-threshold our study is well-powered to reliably detect large effects (Hazard Ratio (HR) > 1.3) the PGSs have on rare endpoints such as hirsutism, and smaller effects (HR > 1.1) on the vast majority of the studied diseases (>1800 cases, 29 traits in both sexes, Supplementary Data 7 and Supplementary Fig. 7). For the cross-sex analyses, we took the sex-specific PGSs, and checked whether these would associate with the studied endpoints in the other sex, using the z-test to compare equality between the original and cross-sex associations. We additionally performed SHBG-adjusted PGS associations to all endpoints to control for the effects of SHBG on total and free T associations. Although in both the PGS and causality analyses we opted for adjusting for the effects of SHBG, BMI, PCOS and menopause in women—(Supplementary Data 7–12 and Supplementary Fig. 10)[42–45]—we remind that related factors may still confound our findings.

Patients and control participants in FinnGen provided informed consent for biobank research, based on the Finnish Biobank Act. Alternatively, older research cohorts, collected prior the start of FinnGen (in August 2017), were collected based on study-specific consents and later transferred to the Finnish biobanks after approval by Valvira, the National Supervisory Authority for Welfare and Health. Recruitment protocols followed the biobank protocols approved by Valvira. The Coordinating Ethics Committee of the Hospital District of Helsinki and Uusimaa (HUS) approved the FinnGen study protocol Nr HUS/990/2017.

The FinnGen study is approved by Finnish Institute for Health and Welfare (THL), approval number THL/2031/6.02.00/2017,

amendments THL/1101/5.05.00/2017, THL/341/6.02.00/2018, THL/2222/6.02.00/2018, THL/283/6.02.00/2019, THL/1721/5.05.00/2019, Digital and population data service agency VRK43431/2017-3, VRK/6909/2018-3, VRK/4415/2019-3 the Social Insurance Institution (KELA) KELA 58/522/2017, KELA 131/522/2018, KELA 70/522/2019, KELA 98/522/2019, and Statistics Finland TK-53-1041-17. The Biobank Access Decisions for FinnGen samples and data utilized in FinnGen Data Freeze 6 include: THL Biobank BB2017_55, BB2017_111, BB2018_19, BB_2018_34, BB_2018_67, BB2018_71, BB2019_7, BB2019_8, BB2019_26, Finnish Red Cross Blood Service Biobank 7.12.2017, Helsinki Biobank HUS/359/2017, Auria Biobank AB17-5154, Biobank Borealis of Northern Finland_2017_1013, Biobank of Eastern Finland 1186/2018, Finnish Clinical Biobank Tampere MH0004, Central Finland Biobank 1-2017, and Terveystalo Biobank STB 2018001.

**Causality analyses and their limitations**. MR analyses treat genetic variants as instrumental variables and their reliability depends on two key assumptions: (1) alleles are randomly assigned, and (2) that alleles that influence exposure do not influence the outcome via any other means. The first assumption is partly controlled by using BOLT-LMM as our model in the primary GWAS analysis together with using PCs to control for population stratification. Although demographic phenomena such as assortative mating (mate choice based on similar characteristics in spouse) may affect effect estimates for measures such as educational attainment, based on within-sibship GWAS[46] biochemical traits such as T levels seem to less affected by such confounders. The second limitation (especially in the form of genetic pleiotropy) is harder to control when using a large number of SNPs as instrumental variables. Susceptibility to confounding by pleiotropy (a gene influences multiple traits via independent biological pathways[47]), overall presents a special issue for genetic studies.

In this case, in line with the observed wide-spread genetic pleiotropy affecting most complex traits, we noted that the GWAS loci contained many genes associated with pleiotropic effects on human phenotypes (for example, *LIN28B*[48], *GCKR*[37] and *TYK2*[49]). The cross-sex analyses further supported the notion that such pleiotropic variants (for example, missense variants in genes like *GCKR*, *SERPINA1*, and *SLCO1B1*, associating with T and SHBG levels but having other biological functions (Supplementary Data 1)[50–52] may partly underlie, e.g., T's correlations with T2D and hypothyroidism.

Therefore, given the vast polygenicity of the studied T traits, we chose latent causal variable (LCV)[53] and MR-Egger[54] as our primary causality analysis methods, designed to take into account pleiotropy-induced confounding. LCV has been proposed to provide more unbiased causality estimates than conventional MR approaches, whereas MR Egger should provide accurate causality estimates under the InSIDE assumption (the genetic variants have pleiotropic effects that are independent in magnitude and are thus not mediated by a single confounder exposure), besides its recommended use as a sensitivity analysis for conventional MR[53,54]. For T traits, we additionally compared the causality results of these models to estimates from CAUSE, a MR method similarly designed to distinguish pleiotropy from true causality (Supplementary Data 2)[55]. In the analysis of other complex traits, for comparison we also ran conventional MR analyses (Inverse-Variance Weighed (IVW)), that remains sensitive for confounding by genetic correlation and pleiotropy[53]. To extend the basic MR Egger analysis and to tease out the potential effects of SHBG on causality estimates of total and free T, we used multivariable MR Egger[56]. LCV reports genetic causality proportion (GCP) as an estimate of causality, under a model where genetic correlation between two traits is mediated by a latent variable having a causal effect on each trait. GCP = 1 means trait 1 is fully correlated with the latent variable, and hence fully causal to trait 2. A high GCP value and a statistically significant effect support partial genetic causality between the traits, and suggest that interventions targeting trait 1 are likely to affect trait 2. The *p* value obtained in the analysis refers to the null hypothesis that the GCP = 0. A highly significant *p* value does not require a high GCP, but in the presence of lower GCP suggests only partial causality between the traits. Positive GCP value indicates causality of trait 1 to trait 2, whereas a negative value indicates support for causality of trait 2 to trait 1. LCV also estimates genetic correlation between the traits. Notably, LCV assumes a single latent variable mediating the effect between two phenotypes, and that the effect can be only unidirectional. Hence it may be confounded by bidirectional causal effects or by the presence of several latent variables. Given these limitations of the methods, despite the vast potential of MR in establishing causal relationships[57], we however propose caution in interpreting these results: also in our case the MR models did not always agree on causality, common to recent MR studies assessing the function of T[23,24,26].

To estimate the how much T could explain sex differences in hemoglobin we calculated $((FT\_m-FT\_f/SD\_FT\_m)*\beta*SD\_H\_m)/((H\_m-H\_f)/SD\_H\_m)$, where FT = mean free Testosterone, m = males, f = females, SD = standard deviation, $\beta$ = MR estimate, H = Hemoglobin in UK Biobank based on ref.[58]. We applied the LCV, MR-Egger, CAUSE, multivariate MR-Egger and IWV models locally using R 4.0.2. The MR analyses were run using TwoSampleMR (v0.5.2)[59] and MendelianRandomization (v0.5.0) R packages[60]. These causality analyses on the FinnGen traits included both autosomal and X-chromosomal SNPs. For all causality analyses X-chromosomal effects in males are presented for (0,2) allele dosage coding. For the traits from public GWAS included in genetic correlation analyses, in 16 out 44 instances we could perform two sample MR (phenotype data not including UK Biobank samples) and in 19/44 instances the studies included X-chromosomal data (Supplementary Data 11). FinnGen represents an independent research cohort from the UK Biobank and thus all FinnGen causality analyses were two-sample MR analyses.

**Reporting summary**. Further information on research design is available in the Nature Portfolio Reporting Summary linked to this article.

## Results

We utilized the rich biochemical and health information available in two population-scale genetic datasets and analysis methods building on GWAS discovery. In brief, we conducted sex-stratified GWAS for T, SHBG and free T based on immunoassay measurements from the UK Biobank, and constructed sex-specific polygenic scores (PGS) for these traits. The PGSs capture the genetic effects on T and SHBG levels, and therefore serve as a proxy for cumulative post-pubertal T exposure. Using an external dataset (Young Finns Study; YFS), we validated the performance of the PGSs. We then investigated the effects of the PGSs on a wide range of diseases across diverse clinical entities using the FinnGen study. Lastly, we evaluated causal relationships and genetic correlations between the studied T traits and complex traits, leveraging publicly available GWAS summary statistics (Fig. 1).

**GWAS and polygenic scores for testosterone traits**. We identified more than a hundred genome-wide significant (*p* < 5e-08) loci the autosomes and the X chromosome combined for all testosterone traits (up to 263 loci for SHBG in males) in the UK Biobank GWAS (Methods; Supplementary Data 1). In both sexes, common variants (allele frequency >1%) covered a large proportion of the trait

variability, with the SNP heritability ($h^2$) estimates ranging from 10% for total T in females to 28% for SHBG in males (Supplementary Data 2). The associated loci were enriched for genes affecting steroid hormone biosynthesis, metabolism and excretion, with preferential expression in the liver for all the studied traits, in line with recent findings (Supplementary Figs. 1 and 2 and Supplementary Data 3)[23–25]. Reflecting the high heritability, T measurements from two different time points were highly correlated in both sexes (Supplementary Fig. 3). Collectively, despite being based on immunoassays which may have limited use in clinical settings especially in females[61,62], these findings illustrate that the UK Biobank data permits construction of robust genetic instruments to study how post-pubertal T and SHBG levels relate to adult health.

The loci affecting SHBG were largely shared between the sexes (genetic correlation ($r_g$) = 0.88, $p = 9.7e-197$), whereas we observed a near-zero genetic correlation estimates for both serum T and free T between males and females ($r_g$ = 0.08 and 0.05, respectively, $p > 0.05$), indicating sex-specific genetic determinants, as previously reported[23,24]. Co-localization analyses between male and female GWASs confirmed the widespread sex-specificity of the genetic loci for T (Supplementary Fig. 4, Supplementary Data 4 and Supplementary Data 5)[24].

Reflecting the sex-specific genetic architecture, we observed strong genetic correlation between total T and SHBG only in males ($r_g$ = 0.78 in males vs. 0.05 in females, Supplementary Fig. 4 and Supplementary Data 2). In females, instead, the genetic determinants for free T were shared with SHBG ($r_g$ = −0.65). Correspondingly, SHBG was found causal for total T levels in males (genetic causality proportion (GCP) = 0.80, $p = 5.8e-05$), whereas in females SHBG appeared to control the free T fraction (GCP = 0.83, $p = 3.8e-07$).

To study the impacts of T and SHBG levels in datasets where these measurements are not directly available, we next constructed sex-specific genetic predictors, PGSs, for each trait applying the LDpred algorithm[39] to the sex-specific GWASs. We tested the predictive ability of the PGSs in the YFS where the phenotypic variance explained by a genome-wide PGS ($R^2$) ranged between 1.1% (male free T) to 9.2% (male SHBG), indicating the PGSs predict T and SHBG levels in an independent cohort (Supplementary Data 6). Notably, the sex-specific PGSs for T and free T had no predictive value in the opposite sex, as expected based on serum T being determined by distinct genetic variants in both sexes (Supplementary Data 6 and Supplementary Fig. 5).

**Studying the links between cumulative T exposure and disease.**
We continued by using the PGS to study how post-pubertal T exposure associates with disease risk, the associations potentially implying causal relationships[28]. In short, these analyses allow for estimating the consequences of having a genetic predisposition to higher or lower T levels (Supplementary Fig. 6). To this end we used the FinnGen data, consisting of 217,464 (94,478 men, 122,986 women) Finnish participants, representing roughly 5% of the Finnish adult population, with genotypes linked to up to 46 years of follow-up within nationwide healthcare registries[41]. We studied 36 diseases with potential links to hormones from the following categories: (1) endocrine and metabolic diagnoses, (2) sex-specific endpoints (many specific to females, e.g., postmenopausal bleeding, (PMB)), (3) cardiovascular and circulatory system, (4) nervous system disease, (5) behavioral and neurological diagnoses and (6) other endpoints like injury risk (Supplementary Data 7). The number of cases ranged from 229 individuals diagnosed with hirsutism to 68,774 statin users. Power analyses supported our ability to detect large effects (HR > 1.3) even for the rarest, and smaller effects (HR > 1.05) on more common phenotypes (Supplementary Fig. 7 and Methods).

We first tested whether the PGSs are associated with disease risk in a sex-specific manner, observing 32 associations ($p < 0.0014$, after Bonferroni correction for 36 independent tests; Supplementary Data 7 and Supplementary Fig. 6). The associations involved endocrine, metabolic and sex-specific disorders, highlighting in particular female-specific endpoints (Fig. 2 and and Supplementary Data 7). In men, both higher total T and SHBG PGSs usually associated with reduced disease risk. In women, both higher total T and free T PGSs generally increased risk for multiple diseases, often showing inverse associations to SHBG.

Underscoring T's and SHBG's involvement in metabolism, larger PGS values for total T and SHBG were associated with reduced T2D risk and statin use in men (for total T in males, HR = 0.94, $p = 1.4e-17$ and 0.96, $p = 1.1e-15$, respectively). In contrast, higher free T PGS in women increased risk for both (HR = 1.09, $p = 1.9e-22$ and HR = 1.02, $p = 0.0011$). Given the shared association profiles with SHBG, we included SHBG as a covariate in complementary analyses to distinguish true T-driven effects. These suggested that associations to metabolic traits were not primarily attributable to androgen action (Supplementary Data 7 and Supplementary Fig. 8). In women, higher T PGS was additionally associated with lower hypothyroidism risk (HR = 0.97, $p = 6.5e-05$), persisting SHBG adjustment (HR = 0.97, $p = 8.6e-05$). We also detected suggestive associations to bone strength and injury risk in both sexes, but with the exception of SHBG and osteoporosis in females (HR = 1.08, $p = 0.00015$), none of these findings survived correction for multiple testing.

In the sex-specific category, we replicated the known associations of T to PCOS and breast cancer risk in women[23] (HR = 1.02, $p = 2.8e-06$ and HR = 1.04, $p = 0.0001$ for free T PGS) (Fig. 2 and Supplementary Data 7). Of the previously unstudied endpoints, we robustly linked T with hirsutism and post-menopausal bleeding (PMB) (HR = 1.45, $p = 2.7e-08$ and HR = 1.05, $p = 0.00032$ for free T PGS), with connections to PCOS, and endometrial cancer, respectively[63,64]. The association to hirsutism (excessive hair growth in a male-type fashion) appeared particularly pronounced, with the risk of the condition almost doubling with a 2 SD change in T. In FinnGen, there was only modest case and genetic overlap between PCOS, hirsutism and PMB cases, potentially reflecting underdiagnosis of PCOS in the dataset[65]. SHBG adjusted analyses further suggested these associations were androgen dependent (Supplementary Data 7 and Supplementary Fig. 8). In contrast, reflecting the close relationship between SHBG and calculated free T in females, some associations for female free T like infertility risk (HR = 1.04, $p = 0.00502$), seemed highly dependent on SHBG, showing generally favorable effects on reproductive traits, e.g., HR = 0.98, $p = 0.00116$ for irregular menstruation).

We observed no statistically significant associations to other diseases (Supplementary Fig. 6), including the 13 neurological/ behavioral endpoints studied, like Alzheimer's disease, alcohol use, and anxiety disorders (all $p > 0.0014$). However, the PGSs did show some nominal clinically-relevant associations: higher PGSs for male free T was linked with prostate cancer and lower osteoporosis incidence (HR = 1.03, $p = 0.0083$ and HR = 0.89, $p = 0.0023$, respectively). Male total T associated with stroke (HR = 0.97, $p = 0.017$), and female total T with cardiac death (HR = 1.04, $p = 0.049$) and Parkinson's (HR = 1.08, $p = 0.017$). Both male free T and female total T PGS showed associations to lower anemia risk (HR = 0.97, $p = 0.042$ HR = 0.98, $p = 0.041$, respectively, Fig. 2 and Supplementary Data 7).

**Evaluation of causal relationships between T and disease.**
Although the PGS associations may imply direct causality of the first trait to another, these are also prone for confounders

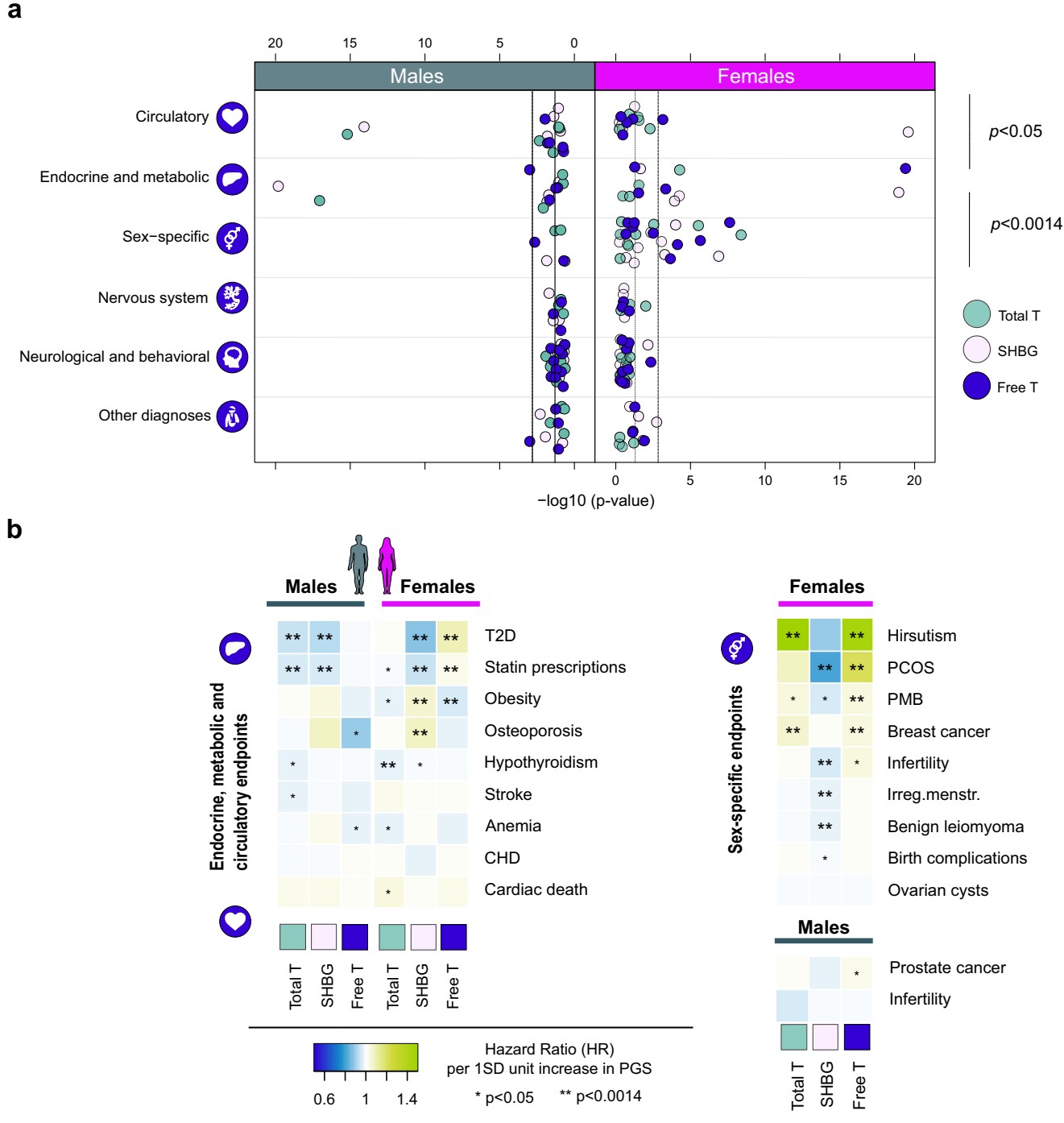

**Fig. 2 Results from the PGS associations with disease endpoints in the FinnGen. a** Illustrates the distribution of association *p*-values by disease category for each biomarker PGS separately for both sexes. For clarity *p* values capped at 1e-20. Dark green symbol = males, pink = females. Green circles = total testosterone, white lilac = SHBG, blue = free testosterone. **b** Shows hazard ratios per one SD increase in PGS for 20 traits from endocrine, metabolic, circulatory and sex-specific categories. Yellow shades = increased risk for endpoint, blue shades = decreased risk for endpoint. The data is based on 94,478 men and 122,986 women from FinnGen R5. Case and control numbers for each endpoint are available from Supplementary Data 7. *\*p < 0.05,* *\*\*p < 0.0014, corresponding to Bonferroni correction for 36 independent traits. PCOS polycystic ovary syndrome, PMB post-menopausal bleeding, CHD coronary heart disease.

including genetic pleiotropy[47]. We estimated causal relationships between T and the studied endpoints using two complementary MR methods that aim to correct for potential pleiotropy: LCV[53], representing a genome-wide approach, and MR-Egger[54] that uses significantly associated SNPs (Methods). For the sex-specific PGS data, in 23/252 instances one or both methods suggested evidence for a causal relationship (p < 0.0014) between a PGS and a disease

(Fig. 3, Supplementary Fig. 9, and Supplementary Data 8). Finally, we distinguished between the effects of T and SHBG by including the latter as a covariate in multivariable MR Egger analyses[56].

Despite the many associations (Fig. 2), also in the MR analyses we observed no clear evidence for T being directly causal for metabolic traits. We consistently saw no significant causality between T and T2D, contrasting some recent findings (Fig. 3, Supplementary Fig. 9,

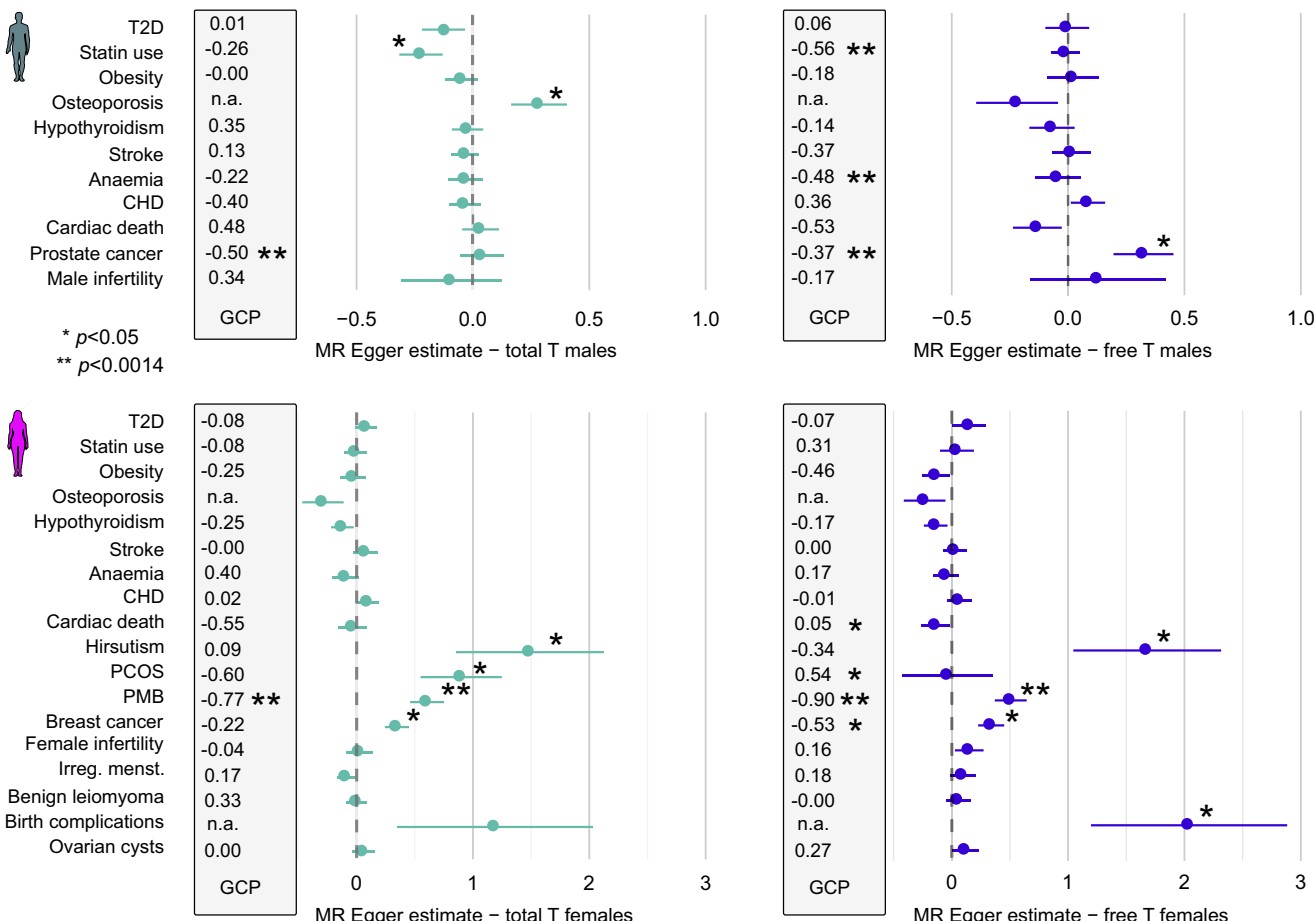

**Fig. 3 Results from causality analyses in FinnGen, showing MR estimates for total and free T in males and females.** The figure includes traits from the endocrine, metabolic, circulatory and sex-specific categories shown in Fig. 2. The data is based on 94,478 men and 122,986 women from FinnGen R5. Case and control numbers for each endpoint are available from Supplementary Data 8. Dark green symbol = males, pink = females. Estimates shown in green = total testosterone, blue = free testosterone. Shown are both genetic causality proportion (GCP) estimates from LCV (within the gray boxes, 0 = no causality, 1 = fully causal, minus sign indicates suggested reverse causality, i.e., the endpoint affecting T), and MR Egger betas and standard error for each trait (horizontal lines). The MR Egger value in the x-axis corresponds to 1 SD increase in risk per 1 SD increase in T/free T. *p < 0.05, **p < 0.0014, n.a. = not applicable due to low heritability estimate in FinnGen under the LCV model.

and Supplementary Data 8)[23]. In addition, the partly surprising links between total T, higher risk for osteoporosis and lower risk for statins seen in standard MR in males appeared attributable to SHBG.

In contrast, the causality analyses strongly supported the role of T in the regulation of female reproductive health. For instance, MR Egger supported causality of total T ($\beta = 0.90$, $p = 0.012$) and LCV of free T to PCOS (GCP = 0.54, $p = 0.0017$) (Fig. 3 and Supplementary Data 8), independently of SHBG. We observed causality also between T, PMB and hirsutism ($\beta = 0.61$, $p = 4.5e{-}05$ and $\beta = 2.11$, $p = 0.002$, respectively, for free T in multivariable MR Egger). Here, adjusting for PCOS had negligible effects on MR results, suggesting these phenotypes are primarily dependent on direct androgen load, and not on metabolic disease (Supplementary Data 8). Additionally, both approaches showed causality between T and hormonal cancers (e.g. GCP = −0.55, $p = 0.041$ and $\beta = 0.34$, $p = 0.0031$ for female free T and breast cancer; GCP = −0.37, $p = 2.8e{-}16$; $\beta = 0.32$, $p = 0.014$ for male free T and prostate cancer). Notably, genetically predicted free T was linked with cancer risk also in the opposite sex (GCP = −0.37, $p = 6.6e{-}19$ for male free T and breast cancer, and GCP = −0.74, $p = 0.0041$ for female free T and prostate cancer).

Lastly, some causal relationships were detected despite no robust associations in the PGS analyses, like the protective association of

female T to seropositive rheumatoid arthritis (RA) (Multivariable MR Egger $\beta = −0.43$, $p = 0.0079$). T and SHBG were linked with injury risk in both sexes (e.g., GCP = 0.53, $p = 0.0018$ and $\beta = 0.17$, $p = 1.3e{-}05$ for SHBG increasing forearm/elbow injuries in men). Finally, LCV suggested a causal relationship between higher SHBG and ADHD in both sexes, and higher free T and increased risk for conduct disorder but decreased risk for emotionally unstable personality in men ($p < 0.0014$, Supplementary Data 8), pointing to potential hormonal involvement in the regulation of neuronal processes.

**Results from the cross-sex PGS associations in FinnGen.** Owing to the unique genetic architecture, the sex-specific PGS for T and free T do not predict the corresponding hormone levels in the opposite sex (Supplementary Data 6). Given this, we reasoned that cross-sex analyses, i.e., analysis of the effect of a sex-specific PGS in the opposite sex, would provide a unique opportunity to assess if the original associations stem from T action, and to detect potential antagonistic effects for the PGSs between the sexes.

Aligning with the results from the MR analyses, 15/20 of nominally significant ($p < 0.05$) sex-specific associations remained similar (Z-test $p > 0.05$) in the cross-sex analyses, pointing to genetic pleiotropy rather than T action as the basis for many metabolic associations. For example, both female and male total T

PGSs associated with hypothyroidism risk with a similar effect size in the opposite sex, and all PGS associations to T2D were replicated in the other sex (Fig. 4 and Supplementary Data 9 and 10).

The associations differed (Z-test $p < 0.05$) in the opposite sex for male total T PGS and stroke, female total T PGS and anemia, and male free T PGS, head injury risk and osteoporosis. (Fig. 4 and Supplementary Data 9 and 10). In addition, the effect of male total T PGS on statin use was attenuated in females. Intriguingly, while higher male total T PGSs rather reduced stroke risk in men (HR = 0.97, $p = 0.017$), this associated with increased risk for stroke in women (HR = 1.03, $p = 0.017$). This was the only endpoint for which we detected evidence for sexual antagonism. The antagonistic effect strengthened with SHBG adjustment (HR = 0.96, $p = 0.011$ in men and HR = 1.07, $p = 0.0010$ in women). The results indicate the genetic effects on stroke risk may be partly sex-specific[66], with potential interplay from sex hormones.

Finally, the cross-sex analyses implied increased androgen load as a direct contributor to poorer reproductive health in women. For the reproductive endpoints that were associated with female free T PGS, i.e., infertility, PMB, PCOS and hirsutism, the male free T PGS had no predictive power (all $p < 0.05$, Fig. 4 and Supplementary Data 9 and 10). Nonetheless, the effect sizes were not attenuated in a statistically significant manner for breast and prostate cancers, ($p > 0.05$, Fig. 4), echoing the results from causality analyses and pointing to a degree of shared genetic risk, irrespective of free T levels, between male and female hormonal cancers.

**Extending the FinnGen discoveries**. We next sought to validate and refine the FinnGen discoveries in additional datasets, extending our analysis to include quantitative traits not available in large numbers in FinnGen. To this end, we used genetic correlation analysis, allowing for estimation of the extent to which two traits share genetic factors[40,67], followed by causality estimations.

We selected 44 traits with publicly available GWAS summary statistics, identical to (e.g., T2D, breast and prostate cancers) or closely reflecting the studied disease phenotypes (heel bone mineral density (HBMD), mood swings) from FinnGen, adding anthropometric traits to the analyses (Supplementary Data 11). For most female-specific phenotypes studied in FinnGen, including hirsutism and PMB, we had no comparable phenotypes, as there were no published GWAS available.

We found evidence of a significant genetic correlation in 72/352 instances ($p < 0.0011$, corresponding to Bonferroni correction for 44 tests) (Fig. 5 and Supplementary Data 11). Reflecting the FinnGen PGS associations, the genetic correlations involved traits related to metabolism, including biomarkers, but we detected only few correlations to behavioral traits, and no significant correlations to neurological or temperamental traits (Fig. 5 and Supplementary Data 11). Notably, the behavioral traits showing significant correlations were clearly linked to metabolism (smoking, sleep duration and exercise).

The genetic factors increasing serum total T and SHBG overall promoted a favorable metabolic profile in males. Despite correlating with increased BMI, total T and SHBG were positively correlated to adiponectin, high-density lipoprotein (HDL) and lower waist-to-hip ratio (WHR) ($r_g > 0.20$, $p < 0.0011$) while lowering triglycerides and T2D risk in men ($r_g < -0.25$, $p < 0.0011$). In contrast, higher free T fraction in women associated with negative metabolic effects, including higher WHR ($r_g = 0.25$, $p = 1.6e-22$) and lower HDL cholesterol ($r_g = -0.18$, $p = 4.2e-05$). Relatedly, strong correlations were observed between SHBG and metabolic traits in women, including protective associations to markers of liver damage (alanine transaminase (ALAT), $r_g = -0.21$, $p = 1.7e-08$) and gamma-glutamyl transferase (GGT), $r_g = -0.16$, $p = 0.0001$).

We observed significant genetic correlations to hormonal cancers in both sexes, as expected[23,26]. In men, the genetic factors increasing free T promoted prostate cancer ($r_g = 0.12$, $p = 0.0004$), whereas in women these increased especially the risk of estrogen receptor (ER) positive breast cancer ($r_g = 0.15$, $p = 9.1e-09$). Finally, consistent with the existing genetic, epidemiological and experimental data[1,26], we found evidence for shared genetic background for T, hemoglobin levels and body fat in both sexes (e.g., $r_g = 0.15$, $p = 1.0e-07$ and $r_g = -0.14$, $p = 0.0002$ for hemoglobin and body fat, respectively, with male free T).

In the LCV and MR-Egger analyses, we found statistically significant evidence ($p < 0.0011$) of a causal relationship in 7% (26/354) of instances across the 44 traits (Fig. 5 and Supplementary Data 12). Reflecting the results from FinnGen, for most traits there was no evidence for substantial causality, but the few suggested causal relationships involved traits with clear biological links to T function. As examples of expected causal relationships[1,26,68], our analyses supported the contribution of free T levels to male-pattern baldness (MPB, GCP = 0.44, $p = 7.7e-19$) and hemoglobin levels (GCP = 0.64, $p = 0.00085$) (Fig. 5 and Supplementary Data 12). In addition, higher T in men was linked to increased number of children fathered (GCP = 0.62, $p = 1.2e-15$ for total T). The MR Egger analyses supported the causality of female T to ER+ breast cancer ($\beta = 0.282$, $p = 0.00014$ for total T), whereas prostate cancer risk was linked to SHBG levels (GCP = $-0.64$, $p = 0.00019$). Additionally, SHBG decreased lymphocyte count in both sexes (GCP = 0.39, $p = 1.6e-07$ for men, GCP = 0.42, $p = 0.00014$ for women), and in men SHBG decreased risk of erectile dysfunction (GCP = 0.42, $p = 0.00032$). Finally, we observed causality between SHBG and age at menopause ($\beta = -0.756$, $p = 0.0010$), suggesting SHBG levels modify reproductive phenotypes across female lifespan.

Overall, we detected relatively little robust statistical evidence for T levels being causal to most traits (Supplementary Data 12), though we noted also some nominally significant associations, like male free T reducing body fat (GCP = 0.50, $p = 0.036$). Yet, emphasizing the intricate relationship between hormone levels and metabolism[69], we for instance detected obesity and liver function related traits being causal for T in females: triglycerides increasing free T levels, (GCP = $-0.80$, $p = 1.4e-06$) and GGT total T levels (GCP-0.60, $p = 4.5e-06$). In combination with the FinnGen data, these results thus further suggest that in most instances the connection between T and SHBG levels and complex traits and diseases is not straightforward.

## Discussion
Since its discovery in the early 20th century, testosterone (T) has been proposed to modify phenotypes and diseases that differ between the sexes, due to the extensive male-female differences in circulating T levels. Besides the biological impacts of normal variation in T levels, the use of T replacement therapy in medical practice has been globally on rise, and the potential risks and benefits associated with T supplementation remain debated[7]. Although highly valuable, clinical trials to investigate the relationship of T levels to many complex phenotypes would be often infeasible or unethical to conduct. Here, using genetic data is an alternative means to estimate how population variability in baseline T levels connects with human health, leading to causal insights beyond the reported epidemiological relationships. In this study we combined data from over 625,000 participants of the FinnGen and UK Biobank cohorts to provide a broad and systematic perspective into the function of long term T exposure, a heritable trait in both sexes that can be proxied by using genetic data, as a regulator of health and disease in men and women. Taken together, supporting recent recommendations, our data suggests that the risks and benefits of using T as a medical

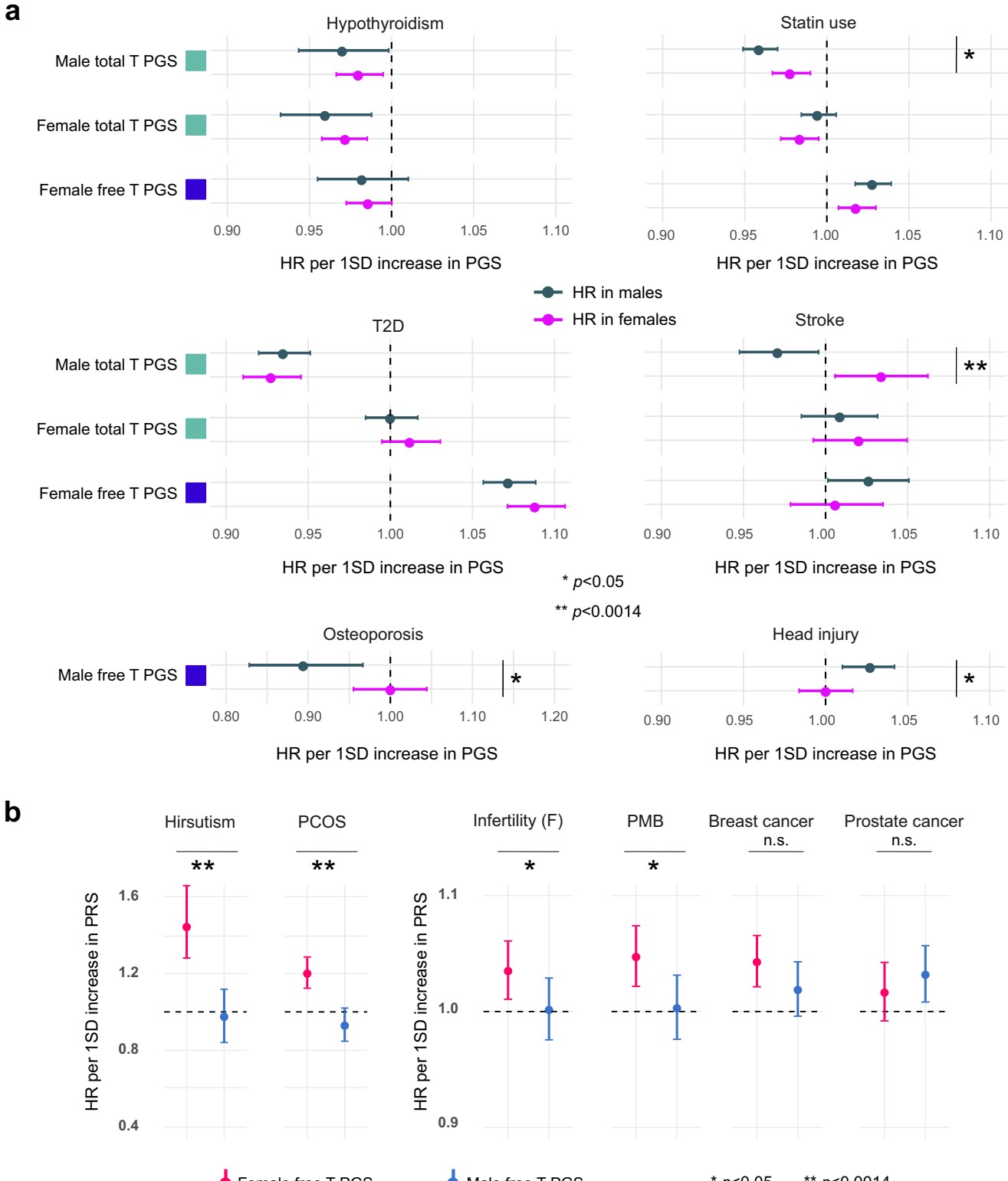

**Fig. 4 Results from the cross-sex PGS analyses. a, b** Show HR point estimates with 95% confidence intervals. **a** Cross-sex PGS associations with hypothyroidism, statin use, T2D and stroke. $p < 0.05$ suggests that the effects of a given PGS vary depending on sex. Green = total T PGS, blue = free T PGS. Dark green = HR in males, pink = HR in females. **b** Illustration of cases with statistical evidence for male and female PGSs having different effects (hirsutism, polycystic ovary syndrome (PCOS), infertility and post-menopausal bleeding (PMB)). No evidence for such difference for breast and prostate cancers. Red estimates = female free T PGS, blue = male free T PGS. The data are based on 94,478 men and 122,986 women from FinnGen R5. Case and control numbers for each endpoint are available from Supplementary Data 9. *Chi-Squared $p < 0.05$, **Chi-Squared $p < 0.0014$, n.s. = not significant.

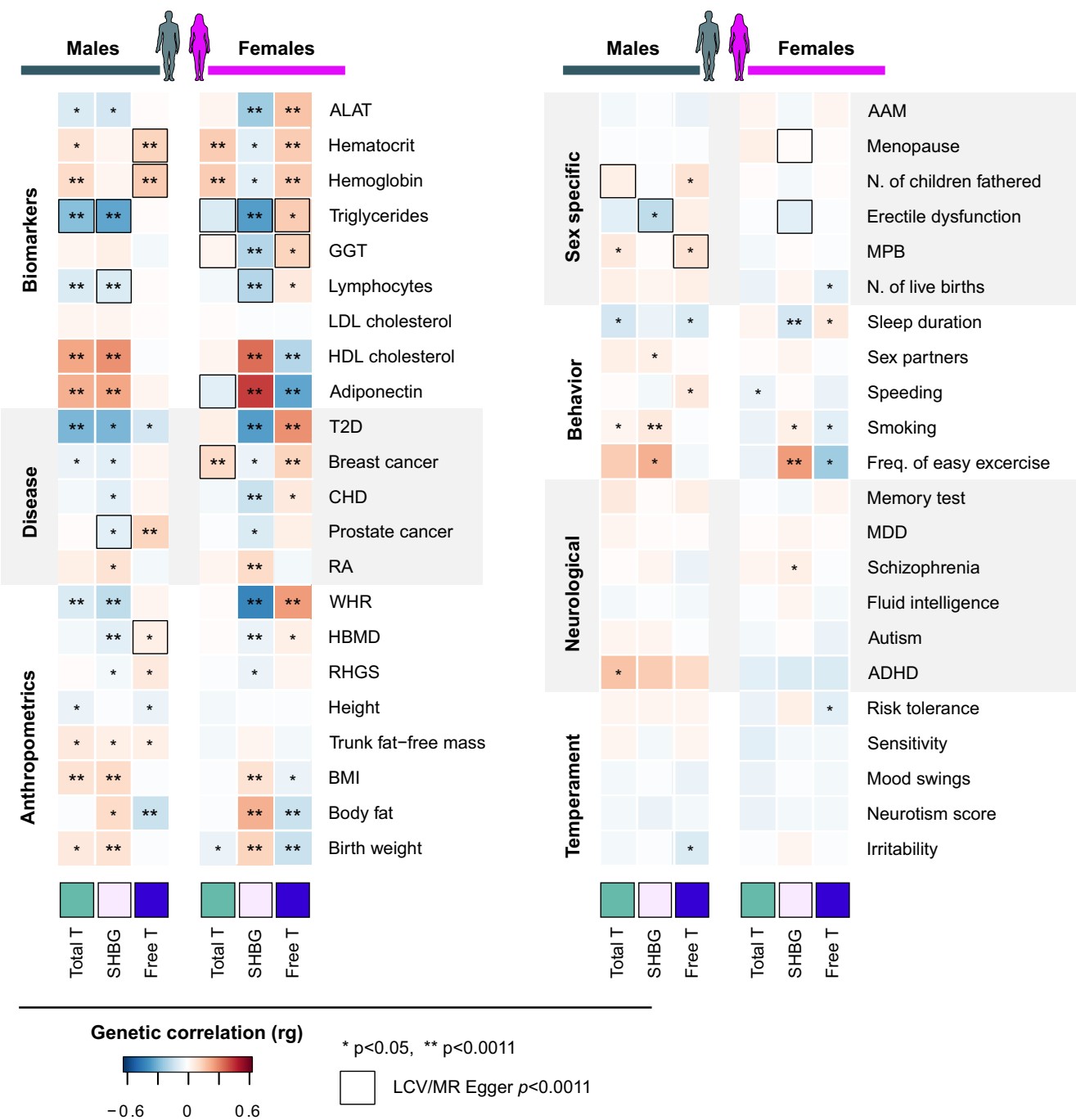

**Fig. 5 Genetic correlation and causal relationships between the 44 traits and phenotypes from public GWAS data based on LDSC and MR analyses.**
The asterisks refer to significant genetic correlation between T/SHBG/free T and the studied phenotypes (*$p < 0.05$ and **$p < 0.0011$, after Bonferroni correction for 44 independent tests). Dark green symbol = males, pink = females. Red shades in boxes refer to positive genetic correlation (the same genetic factors increase both traits), and blue shades to negative genetic correlation (genetic factors increasing one trait decrease the other). Genetic correlation does not necessarily imply causality. Black squares mark significant causal relationships between the tested traits based on LCV and MR Egger analyses. Data is based on GWAS summary statistics from studies ranging from 18,759 MDD cases to over a million participants in risk tolerance GWASs, most studies including several hundreds of thousands of subjects (Supplementary Data 11). Green = total testosterone, white lilac = SHBG, blue = free testosterone. AAM age at menarche, ALAT alanine transaminase, ADHD attention deficit and hyperactivity disorder, CHD coronary heart disease, GGT gamma-glutamyl transferase, HBMD heel bone mineral density, MDD major depressive disorder, MPB male pattern baldness, RA rheumatoid arthritis, RHGS relative handgrip strength, WHR waist-to-hip ratio.

treatment should be carefully weighted, given T's complex and indirect relationship to most phenotypes and potential adverse and beneficial outcomes in both sexes[7,26].

We observed three major themes regarding genetically determined T's contribution to disease. First, we detected causality

between higher T levels and several sex-biased phenotypes with biological links to T, but less contribution to most other phenotypes, echoing experimental data and findings from recent MR studies[23,26]. Whereas our analyses supported causality of genetically determined T, e.g., to hormone-sensitive cancers and

hirsutism, this did not apply for traits such as obesity, T2D or hypothyroidism. Accordingly, we stress the complex relationship of T levels with many metabolic and endocrine traits. The results echo the epidemiological observations that lower T levels correlate with increased risk of metabolic disease in men, while in women the opposite hold true[70,71]. However, our data shows that these associations to metabolic health may reflect widespread genetic pleiotropy, implying complex etiology as opposed to T action. We further estimate that normal heritable variation in T levels—contrary to popular beliefs—has only modest effects on many phenotypes. In the light of our study, despite the potential behavioral impact of hormonal processes, for example the concept of explaining the risk for many temperamental and neurological phenotypes with heritable differences in adult T levels[72,73] appears unsubstantiated.

Secondly, we highlight how our data reflects the correlations between total T, SHBG and free T levels in both sexes. Total T in men correlates with SHBG levels epidemiologically[13–15] and at the genetic level, meaning that same genetic variants affect both traits. Underscoring the contribution of SHBG to total T associations, we found SHBG causal for total T levels in males. Especially for many metabolic traits, SHBG—either directly or through its metabolic network—appeared to modify some of total T's associations and causality estimates. Consistent with potentially divergent biological effects for the SHBG bound and unbound T, we however observed distinct association profiles for total and free T fractions in men. For women, the situation was the opposite: whereas total T associations seemed largely independent of SHBG action, for many metabolic traits the high dependency of calculated free T on SHBG levels was evident.

Finally, we report that genetic differences affecting T levels seemed to affect disease risk especially in women, including PCOS-related endpoints like hirsutism, PMB and infertility. Although our study suggests that genetically determined T and PCOS may be partly different entities, we unfortunately cannot state if genetically determined T has fully PCOS-independent effects on these traits, since our study may be limited by PCOS being potentially underdiagnosed in FinnGen, or the diagnosed PCOS cases representing only a subpopulation of this heterogenous disorder[74,75]. In addition, all these traits correlate with obesity[32], that in turn correlates with PCOS and T levels (Supplementary Fig. 10 and Supplementary Data 8), and we may have not been able to fully account for such confounding.

Out of the metabolic traits, our T2D results that are in line with clinical trials but contrast recent genetic findings[23,33,76] merit an extended discussion. We note that some of our MR analyses also supported a causal link between T and T2D, like previously shown[23,24]. However, the MR models developed to address confounding by genetic pleiotropy, including variants effects on SHBG, did not support causality. Finally, the cross-sex analyses, relying on the T PGSs not predicting T levels in the opposite sex, strongly suggested that the T/T2D associations do not reflect causality. For both male and female specific PGSs, the T/T2D associations remained similar also in the other sex, meaning that the original sex-specific associations unlikely stem from T action. Since we detected limited evidence also for SHBG's causality to T2D, echoing some recent findings[77], our data thus suggests that overall genetic pleiotropy and factors affecting, e.g., BMI or liver function, may be more likely accountable for the connection between T levels and T2D.

Having comprehensively mapped the impacts of T across diverse complex disease and traits, we continued to speculate on the role of T levels as a contributor to the male-female differences. Overall, higher T levels associated with typical male characteristics. For instance, our work suggested T is causally connected to higher hemoglobin and bone strength and lower body fat, backed up by previous experimental observations[1,78,79]. Extrapolating from the MR results, we estimate that ~10–20% of the mean difference in hemoglobin levels between men and women may result from average differences in free T levels, consistent with the notion that T directly affects male-female differences in, e.g., athletic capacity[1]. Moreover, higher T levels were found causal to masculine external features like hirsutism and baldness. The fact that T levels only partly explain these traits and the limited evidence for the causal involvement of T for many traits showing sex differences like cardiovascular disease however suggests that sex differences in disease prevalence are not attributable to a linear relationship between T levels and a phenotype. Considering the 7–15× differences in T levels between an average male and female, that should lead to vast differences in disease incidence between the sexes if adult T levels were a major determinant for health, this is only to be expected.

We however provide some unique insight into the genetic causes behind the sex differences in T levels. Cross-sex genetic correlations for traits related to fitness (e.g., reproductive success) are generally expected to be low, due to potentially conflicting evolutionary pressures[80]. We indeed associated T positively with number of offspring in men, but negatively with reproductive health in both pre- and postmenopausal women. Besides the well-established biological dichotomy between the sexes, leading to vast differences in T levels, we may speculate that there exists a selective advantage in relation to reproductive success to maintain adequate T levels in males, whereas the opposite may be true for females. This in turn might partly promote the widespread sex differences across multiple traits, and the unique sex-specificity of genetics of T. In the cross-sex PGS analyses of disease endpoints common to both sexes, we however found only one case with clear evidence of antagonistic associations for a T PGS. The male-specific total T PGS protected from stroke in men, whereas the same PGS had opposite effects on stroke risk in women. Although we cannot conclude that the action of T drives these associations, the result agrees with sex-specificity in the genetic disease mechanisms for stroke[66].

Although based on extensive data sets, our study has limitations. The UK Biobank data is based immunoassays, which may have limited use in clinical settings. Yet we show that the T measures from two distinct visits are highly correlated in both sexes, which together with the robust genetic findings illustrates that the data allows for estimating how variable T levels relate to adult health. We however stress that our genetic data clearly suggests that many different biological processes are involved in determining normal variation in T levels, serum T levels likely serving only as a proxy for T action in tissues. Furthermore, although the PGS for total T and SHBG, (based on participants aged between 39–72 years, women mostly post-menopausal) explained a substantial fraction of the trait variance also in younger individuals ($R^2$ 9.2% for SHBG in males in the YFS), for example male free T PGS seemed to yield less accurate predictions (1.1% in YFS). Since the majority of the variance remains uncaptured by the PGSs, we cannot draw conclusions about the function of T based on the PGSs alone, and results may not be fully generalizable to other than European populations. Nonetheless, we seem to have been well powered to detect any substantial effects for the PGS (Methods and Supplementary Fig. 7). In line with the data from other complex traits, it thus seems that for most persons information about T PGSs may have limited use, but for the extremes of distribution such information may prove clinically valuable in the future[81].

Given the combination of analysis strategies—for the MR estimates, 1 SD change in circulating T levels as the basic unit—

we were well positioned to detect any substantial effects normal variation in adult T levels, excluding clinical conditions, would have on disease and health. Yet we remind that the MR methods have some important limitations, including susceptibility to genetic pleiotropy (Methods), and these did not always agree upon causality. We further note that the emphasis of our study has been on adult T levels, whereas some of the effects of T might be organizational, acting during specific developmental windows. Our setting thus may not allow for assessing the effects of fetal T exposure, which may be critical, e.g., for neurological traits[82], or serum T may not be the optimal androgen to assess such risk associations. We also emphasize that our results are based on normal, heritable variation in T levels, not, e.g., on supraphysiological synthetic T use wrecking baseline hormonal balance, which, combined with specific external stimuli such as exercise regime are linked with more drastic effects, e.g., on muscle strength and health[83]. Additionally, many of T's effects depend on its conversion to estradiol also in males, and we cannot rule this out as a potential confounder in our study. Finally, the data used in our study does not allow for assessing the effects of acute changes in hormone secretion, including those happening during different phases of the menstrual cycle in females or upon certain environmental stimuli, and personal differences in responses to such fluctuations.

Despite these challenges, in this study we highlight several previously unreported, albeit often expected relationships with genetically determined T levels, human health, and sex differences. Besides the gained medical insight, underscoring some critical factors that should be considered when assessing these connections, our study therefore provides a novel reference point for future genetic and epidemiological studies on T.

## Data availability

The source data for figures is available in Supplementary Data: for Fig. 1—Supplementary Data 2; Fig. 2—Supplementary Data 7; Fig. 3—Supplementary Data 8; Fig. 4—Supplementary Data 9; Fig. 5—Supplementary Data 10 and 11. Full genetic and clinical data from FinnGen and the UK Biobank are available for researchers by application, https://www.finngen.fi/en/access_results and https://www.ukbiobank.ac.uk/. For YFS, see https://youngfinnsstudy.utu.fi/index.html. GWAS summary statistics for total T, SHBG, FAI and free T based on the UK Biobank data will be available from the GWAS catalog (https://www.ebi.ac.uk/gwas) with submission ID GCP000476. Data from publicly available GWASs can be downloaded from the source repositories with references listed in Supplementary Data 11.

## Code availability

The full genotyping and imputation protocol for FinnGen is described at https://doi.org/10.17504/protocols.io.nmndc5e[84].

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

## Acknowledgements

The research has been conducted using the UK Biobank Resource under application number 22627. The FinnGen project is funded by two grants from Business Finland (HUS 4685/31/2016 and UH 4386/31/2016) and eleven industry partners (AbbVie Inc, AstraZeneca UK Ltd, Biogen MA Inc, Celgene Corporation, Celgene International II Sarl, Genentech Inc, Merck Sharp & Dohme Corp, Pfizer Inc., GlaxoSmithKline, Sanofi, Maze Therapeutics Inc., Janssen Biotech Inc). The funders had no role in study design, data collection and analysis, decision to publish or preparation of the manuscript. The following biobanks are acknowledged for collecting the FinnGen project samples: Auria Biobank (https://www.auria.fi/biopankki), THL Biobank (https://thl.fi/fi/web/thl-biopankki), Helsinki Biobank (https://www.terveyskyla.fi/helsinginbiopankki), Biobank Borealis of Northern Finland (https://www.oulu.fi/university/node/38474), Finnish Clinical Biobank Tampere (https://www.tays.fi/en-US/Research_and_development/Finnish_Clinical_Biobank_Tampere), Biobank of Eastern Finland (https://ita-suomenbiopankki.fi), Central Finland Biobank (https://www.ksshp.fi/fi-FI/Potilaalle/Biopankki), Finnish Red Cross Blood Service Biobank (https://www.veripalvelu.fi/verenluovutus/biopankkitoiminta) and Terveystalo Biobank (https://www.terveystalo.com/fi/Yritystietoa/Terveystalo-Biopankki/Biopankki/). All Finnish Biobanks are members of BBMRI.fi infrastructure (www.bbmri.fi). The Young Finns Study has been financially supported by the Academy of Finland: grants 322098 (T.L), 286284, 134309 (Eye), 126925, 121584, 124282, 129378 (Salve), 117787 (Gendi), and 41071 (Skidi); the Social Insurance Institution of Finland; Competitive State Research Financing of the

Expert Responsibility area of Kuopio, Tampere and Turku University Hospitals (grant X51001); Juho Vainio Foundation; Paavo Nurmi Foundation; Finnish Foundation for Cardiovascular Research; Finnish Cultural Foundation; The Sigrid Juselius Foundation; Tampere Tuberculosis Foundation; Emil Aaltonen Foundation; Yrjo Jahnsson Foundation; Signe and Ane Gyllenberg Foundation; Diabetes Research Foundation of Finnish Diabetes Association; This project has received funding from the European Union's Horizon 2020 research and innovation program under grant agreements No 848146 for To Aition and grant agreement 755320 for TAXINOMISIS; European Research Council (grant 742927 for MULTIEPIGEN project); Tampere University Hospital Supporting Foundation and Finnish Society of Clinical Chemistry. We greatly thank all UK Biobank, the Young Finns study, and FinnGen participants, as well as the principal investigators, laboratory personnel and data management teams behind these efforts. This work was supported by the Academy of Finland (https://www.aka.fi/en/) grants 331671 to N.M., 312072 to T. Tuomi, 288509 and 319181 to M.P., and 315589 and 320129 to T. Tukiainen, and Academy of Finland Center of Excellence in Complex Disease Genetics grants 312075 to M.D., 312062 and 336820 to S.Ripatti and 312076 to M.P.), by the Finnish Foundation for Cardiovascular Research (https://www.sydantutkimussaatio.fi/en) (S.Ripatti), the Sigrid Juselius Foundation (https://sigridjuselius.fi/en/) (S.Ripatti, M.P., T. Tukiainen) and University of Helsinki (https://www.helsinki.fi/en) HiLIFE Fellow and Grand Challenge grants (S.R. and M.P.), and 3-year research project grant (T. Tukiainen).

## Author contributions

J.T.L. and T. Tukiainen conceptualized the study, designed the analysis plan and wrote the manuscript. N.M. calculated the PGSs for FinnGen and performed PGS associations. L.E.L. performed GWASs in the UK Biobank. A.A.-O. calculated PGSs and assisted in the analysis of YFS. S. Ruotsalainen performed GWAS in FinnGen, if desired summary statistics were not directly available. J.T.L. and T. Tukiainen performed the rest of the analyses. T.L., M.K., O.R. and FinnGen consortium provided data, T.P. and T. Tuomi clinical and M.P. statistical expertise. N.M., T.P., T. Tuomi, M.D., S. Ripatti and M.P. provided insightful comments on the study and text improving the manuscript. All authors participated in writing and reviewing the manuscript, and approved the final version of the manuscript.

## Competing interests

The authors declare no competing interests.

## Additional information

## FinnGen Consortium

Nina Mars[17], Sanni Ruotsalainen[17], Mika Kähönen[5,18], Terhi Piltonen[19], Tiinamaija Tuomi[17,20,21,22,23], Mark Daly[17,24], Samuli Ripatti[17,24,25] & Taru Tukiainen[17]

[17]Institute for Molecular Medicine Finland (FIMM), HiLIFE, University of Helsinki, Helsinki, Finland. [18]Faculty of Medicine and Health Technology, Tampere University, Tampere, Finland. [19]Department of Obstetrics and Gynecology, PEDEGO Research Unit, Medical Research Centre, Oulu University Hospital, University of Oulu, Oulu, Finland. [20]Department of Endocrinology, Abdominal Centre, Helsinki University Hospital, Helsinki, Finland. [21]Folkhalsan Research Center, Helsinki, Finland. [22]Research Programs Unit, Clinical and Molecular Metabolism, University of Helsinki, Helsinki, Finland. [23]Lund University Diabetes Centre, Department of Clinical Sciences Malmö, Lund University, Malmö, Sweden. [24]Broad Institute of MIT and Harvard, Cambridge, MA, USA. [25]Department of Public Health, Faculty of Medicine, University of Helsinki, Helsinki, Finland. A full list of members and their affiliations appears in the Supplementary Information.

