## [Peer Review File · Communications Medicine]

This manuscript has been previously reviewed at another Nature Portfolio journal. This document only contains reviewer comments and rebuttal letters for versions considered at Communications Medicine

Reviewers' comments:

Reviewer #2 (Remarks to the Author):

Thank you for an incredibly detailed response! In particular, I think it is much clearer which analyses used autosomal variants and this is much better justified. My questions have been addressed very thoroughly and I have no further comments.

Reviewer #3 (Remarks to the Author):

In submitting this manuscript, the authors have responded to reviews from submission to a previous journal. They revised the manuscript taking account of these comments. They addressed all of the comments from the previous reviews and substantially improved the manuscript. I have no further comments.

Reviewer #4 (Remarks to the Author):

Comment 1

The novel contributions of the study are still not clearly articulated. Given the limitations of the genetic analyses to assess the impact of variation in circulating T levels in the physiologic range (now addressed in the response to Comment 3), the study provides incremental information on the biology of T. Further, the association of T with hirsutism and infertility, in addition to PCOS, likely reflects the fact that PCOS is the most common cause of hirsutism and anovulatory infertility. Women with PCOS also have an increased prevalence of obesity, which is associated with PMB. Thus, genetically determined T is likely associated with only one female reproductive trait, PCOS, analogous to the findings of Ruth et al. This information should be added to the manuscript.

Comment 2

The manuscript lacks an in-depth discussion of the relevance of the findings to the field. The discrepant Mendelian randomization results for genetically determined T and type 2 diabetes risk in the current manuscript and the manuscript of Ruth et al merit a more comprehensive explanation. The possibility that previous Mendelian randomization analyses were limited by horizontal pleiotropy should be addressed.

The authors refer to SHBG as a “potential confounder” of phenotypes associated with T. However, there are now a number of Mendelian randomization studies implicating genetically-determined SHBG in the pathogenesis of type 2 diabetes. The mechanisms for this association remain unknown but it is entirely possible that SHBG has biological actions in addition to its role as a transport protein for sex steroids. Further, the authors analyze free T and FAI as independent endpoints when both are derived parameters calculated from total T and SHBG levels. The analysis should be limited to T and SHBG. The independent role of SHBG on endpoints should be considered.

Comment 3

Genetically-determined T accounts for lifelong exposure so it will reflect both organizational and activational T actions. There is an extensive body of experimental data on T administration in humans that precisely characterizes dose-response effects of T within the physiologic range on a variety of relevant, accurately assessed endpoints. Given the limitations of the genetic analyses that the authors now acknowledge, lines 404-419, the manuscript provides limited insight into the contribution of T to disease risk and complex traits.

Comment 4

The associations of genetically-determined T with PCOS likely account for its association with hirsutism, infertility and PMB, all features of PCOS.

Response to reviewer's comments

This document provides detailed answers to all the questions raised by the reviewers. We thank the reviewers for taking the time to help us in this process, and especially for the insightful comments that have led to improvement of our manuscript. We have addressed all the comments also in the second review round, and modified our manuscript accordingly.

The original reviewer comments are presented *in italics with grey font*, our replies in black, normal type font, and **the changes made to the manuscript with red font**.

Reviewer #2:

Thank you for an incredibly detailed response! In particular, I think it is much clearer which analyses used autosomal variants and this is much better justified. My questions have been addressed very thoroughly and I have no further comments.

Author reply: We are very happy to hear these comments and thank the reviewer once more for taking the time to help us to improve our manuscript.

Reviewer #3:

In submitting this manuscript, the authors have responded to reviews from submission to a previous journal. They revised the manuscript taking account of these comments. They addressed all of the comments from the previous reviews and substantially improved the manuscript. I have no further comments.

Author reply: We are very happy to hear these comments and thank the reviewer once more for taking the time to help us to improve our manuscript.

Reviewer #4:

Comment 1

The novel contributions of the study are still not clearly articulated. Given the limitations of the genetic analyses to assess the impact of variation in circulating T levels in the physiologic range (now addressed in the response to Comment 3), the study provides incremental information on the biology of T. Further, the association of T with hirsutism and infertility, in addition to PCOS, likely reflects the fact that PCOS is the most common cause of hirsutism and anovulatory infertility. Women with PCOS also have an increased prevalence of obesity, which is associated with PMB. Thus, genetically determined T is likely associated with only one female reproductive trait, PCOS, analogous to the findings of Ruth et al. This information should be added to the manuscript.

Author reply: It is very unfortunate that despite our best efforts we still have not been able to emphasize the significance and novelty of our article to all readers. Like stated already in our previous responses and in our manuscript, we do acknowledge that the biology of testosterone has been intensely studied, and that some genetic papers on this topic have recently been published. However, we strongly feel that the set of analyses presented in our paper considerably extends the previous literature, and even challenges some of the existing concepts.

We understand the concern of the reviewer that some of our findings might just reflect the variable phenotypes associated with PCOS. With our data we can actually demonstrate that this is not the case. For example, although a common perception is that hirsutism in females (biologically, and now also genetically associated with higher testosterone levels in women in our study) is usually a consequence of PCOS, our data shows that predisposition to clinically diagnosed hirsutism may be different from predisposition to PCOS.

The evidence is threefold. 1) Only a small proportion of hirsutism cases in the FinnGen data have PCOS, and vice versa. 17.6% of hirsutism diagnoses in FinnGen R5 have comorbidity with PCOS, whereas the

prevalence of PCOS cases with hirsutism diagnosis remains only at 6.5%. These numbers are lower than often cited in the literature, and likely reflect the fact that the hirsutism diagnoses in FinnGen, that come from hospital records, may capture only the more severe end of female excessive hair growth, i.e., hirsutism diagnoses are primarily given to those individuals who seek medical attention primarily for this cause. Overall, the hirsutism diagnoses in our study thus seem largely independent from PCOS. Thanks to the comment we now explicitly mention this in the manuscript in several places.

2) Genetically, the FinnGen hirsutism endpoint and PCOS are only moderately and non-significantly correlated ($r_g = 0.49$ (0.47), $p=0.29$, LDSC). This further supports the view that although some common background may exist, the genetic factors predisposing to these traits are largely independent, and we are dealing with two distinct diagnoses.

3) When we compare how our Mendelian Randomization (MR) results change upon adjusting the effect of testosterone SNPs with their effects on PCOS, there is very little support for the concept that PCOS mediates the causality of testosterone to hirsutism. In regular MR analyses we see evidence of a causal relationship between genetically determined T levels and hirsutism (for example, $\beta = 1.61$ (0.34), $p=2.8e-06$ in IVW), and multivariable MR analyses confirm this association is not mediated by PCOS (after adjusting by the SNPs effects on PCOS: $\beta = 1.59$ (0.35), $p=7.0e-06$, results now presented in Supplementary table 22.)

What comes to the suggestion that testosterone's associations to PMB might be explained largely by PCOS-related obesity, there is similarly little evidence to support this claim. For example, the comorbidity between PCOS and PMB in our dataset is extremely small (1.1%) with only 7 cases identified in FinnGen R5 in our analyses with both these diagnoses. Although it is likely that some shared genetic background exists between these traits, and PCOS is likely underdiagnosed especially in older individuals in our data, we detect only weak evidence of these traits being genetically partly overlapping ($r_g = 0.63$ (0.35), $p=0.078$). Additionally, similarly to the case presented above, T's causal effect on PMB does not change upon adjustment of T SNPs effects either on PCOS or obesity (Original effect: $\beta=0.332$, $p=6.1e-05$ vs

beta=0.315, $p=1.2e-04$ (PCOS-adjusted) and beta=0.341, $p=3.8e-05$ (Obesity-adjusted), Supplementary table 22). Hence, we conclude that in contrast to the suggestion by the reviewer, PMB is another trait that seems independently related to T levels, rather than being just a consequence of PCOS or obesity. On the contrary, underscoring the complexity of PCOS-linked phenotypes, as opposed to hirsutism and PMB, our data supports for example the idea that the female infertility-endpoint that the reviewer also highlights seems much more strongly related to obesity and SHBG than T. For example, genetic correlation (r_g) of obesity vs. female infertility = 0.38 (0.16), $p=0.020$ and causality of SHBG to female infertility $p<0.05$ in both LCV and MV-IVW analyses.

In summary, we are grateful to the reviewer for these comments. Whilst preparing our response and making changes to the manuscript, we feel the comment has pointed us to highlight novel aspects that - partly accountable to the wide range of analyses and results in our study - were not emphasized enough in the previous round. In the current version of the manuscript, we hope to have made these points more clear.

Among the other novel discoveries presented in our manuscript, we now emphasize the fact that many of our findings related to reproductive traits are unrelated to PCOS in the abstract, the main text and in the supplementary files.

Based on this comment 1 and also based on comment 4, we have made the following changes into our manuscript:

Abstract, page 1 rows 22-24: "We find genetically predicted T affects sex-biased and sex-specific traits, with a particularly pronounced impact on female reproductive health across lifespan, including causal contribution to traits like hirsutism and post-menopausal bleeding (PMB), **independent of PCOS**"

Results, page 7, rows 163-166: "With the exception of PCOS, total T and free T displayed similar associations to these traits. **In FinnGen, there was modest case and genetic overlap between PCOS, hirsutism and PMB, suggesting the associations of T to latter phenotypes are not PCOS driven**

(Supplementary Table 21). SHBG adjusted analyses further suggested these associations were strictly androgen dependent (Supplementary Table 21, Supplementary Table 22 and Supplementary Figure 6). “

Results page 10, rows 205-208: Under both MR Egger models, we observed causality between total and free T and PMB and hirsutism ($\beta=0.61$, $p=4.5e-05$ and $\beta=2.11$, $p=0.002$, respectively, for free T in multivariable MR Egger). These effects did not change upon adjusting for the effects of T SNPs on PCOS, further suggesting these phenotypes are primarily androgen driven, and not just simply a consequence of PCOS (Supplementary Table 22). At the same time, MR Egger rather supported causality of total T ($\beta=0.90$, $p=0.012$) and LCV of free T to PCOS (GCP=0.54, $p=0.0017$).“

Discussion, page 18, rows 350-352: Based on our analyses, three major themes emerged regarding T's contribution to disease. First, we report that the studied PGSs associated with disease risk especially in females, including many sex-specific endpoints like hirsutism, PMB and infertility. Importantly, we also show that these effects do not depend on the known link between T levels and PCOS.

Additionally:

Supplementary Table 21 now includes a sheet with a table showing case overlap and genetic correlations between PCOS and related traits in FinnGen R5.

Supplementary Table 22 now includes a sheet with tables containing results on multivariable MR analyses. Here we have used information from the SNPs effects on PCOS and obesity risk to test which causal effects remain for female total and free T, after adjusting for these two conditions.

Comment 2

The manuscript lacks an in-depth discussion of the relevance of the findings to the field. The discrepant Mendelian randomization results for genetically determine T and type 2 diabetes risk in the current manuscript and the manuscript of Ruth et al merit a more comprehensive explanation. The possibility that previous Mendelian randomization analyses were limited by horizontal pleiotropy should be addressed.

The authors refer to SHBG as a “potential confounder” of phenotypes associated with T. However, there are now a number of Mendelian randomization studies implicating genetically-determined SHBG in the

pathogenesis of type 2 diabetes. The mechanisms for this association remain unknown but it is entirely possible that SHBG has biological actions in addition to its role as a transport protein for sex steroids. Further, the authors analyze free T and FAI as independent endpoints when both are derived parameters calculated from total T and SHBG levels. The analysis should be limited to T and SHBG. The independent role of SHBG on endpoints should be considered.

Author reply: We thank the reviewer for the suggestion to highlight and elaborate our T2D finding more. It is, indeed, one of the major findings our genetic analyses provide over Ruth. et al 2020, and represents an important result in terms of the ongoing debate over T's role in T2D. In our manuscript, we highlighted the finding by presenting ample evidence for the lack of causality between T levels and T2D (i.e. MR results and cross-sex PGS associations), and discussed some potential reasons for this result, including genetic pleiotropy, and confounding by SHBG. **In the current version of the manuscript, given the comment of the reviewer, we have emphasized this finding and explained the possible reasons behind the discrepancy to the Ruth et al result, adding an extended paragraph discussing these findings to the manuscript.**

On top of this issue, the reviewer brings forward the important and independent role of SHBG. We have included SHBG either as an independent factor or covariate in many of our analyses, to tease out the SHBG-independent effects of T to complex traits and disease. **We now discuss the MR findings and potential SHBG-related confounding in more detail, and for example now clearly acknowledge that we cannot fully exclude causality by SHBG explaining, for example, a part of the T's T2D associations.**

Finally, the reviewer also questions our choice of including FAI and free T as independent phenotypes in our analyses. We agree with the reviewer that these are partly overlapping (and calculated, not directly measured) phenotypes with T and SHBG. However, we argue that these phenotypes bring additional information beyond T and SHBG and hence feel that all of these phenotypes should be included in the study. First, there are already many papers in the field that have used either FAI or free T as a phenotype, including a genetic analysis of the impacts of free T on health-related traits in men (for example, DOI: 10.7554/eLife.58914), and therefore presenting results for these phenotypes will allow for comparisons

between studies for interested readers. Secondly, similarly to the relationship of T and T2D, there is an ongoing debate over whether the biologically active fraction of T (proxied by free T and FAI) has different effects on human phenotypes than total serum T. In fact, our results show that especially in males these two fractions have different association profiles and the analysis of free T as an individual trait provides us additional information beyond focusing on T and SHBG individually or using these in the same model. For example, we detected the strongest causal links between hemoglobin-related traits (including anemia), male-pattern baldness, and prostate cancer with free T in males, whereas such strong links were not detected for total T. Yet, we admit that by analyzing four separate traits (total T, SHBG, FAI and free T) we add an extra layer of complexity to our work.

In response to this comment, we conclude that the inclusion of results for all these four traits all allows for better comparison of our results to existing literature, and secondly also brings novel information e.g. in the form of shown distinct association profiles of total T and free T in males. Hence, we've chosen to maintain information on all these traits in our manuscript also in this second round of revision. **We have now clarified the relationship between these endpoints, and for example state in the discussion clearly that these are calculated endpoints, which reflect the complex nature of T and SHBG relationship.**

Based on these comments, we have made the following changes into our manuscript:

Discussion, page 18, rows 352-355:

Secondly, we highlight distinct association profiles for total T and free T especially in males (**free T calculated based on T, SHBG and albumin levels**), consistent with proposed divergent biological effects for the bound and unbound T fractions. This underscores the potential role of SHBG as a confounder, **besides general genetic pleiotropy**, on many phenotypes associated with T (13-15).

Discussion, page 22-23, rows 436-456:

“When based on a large number of variants, the MR analyses often include variants that do not fulfill the strict definition of an instrumental variable, **which is an essential criterion for most conventional MR**

analyses including inverse-variance-weighted (IVW) MR. Due to the presence of such variants in the data, we made a conscious decision to concentrate on results from MR models addressing pleiotropy. Here we show via MR and cross-sex PGS analyses that such pleiotropic variants (for example, missense variants in genes like *GCKR*, *SERPINA1*, and *SLCO1B1*, affecting T and SHBG levels but also having other established biological functions (Supplementary Tables 1-8)(52-54) may partly underlie for example T's associations with T2D and hypothyroidism.

Especially our T2D results that are in line with previous clinical trials but contrast recent genetic findings (22, 52, 53) merit an extended discussion. We note that also in our case the IVW MR analyses generally supported a causal link between T and T2D, like previously reported (22, 23). However, the MR models developed to address pleiotropy (MR Egger, LCV and multivariable MR) did not support significant causality between any of these traits. Here, the fact that T SNPs do not predict T levels in the opposite sex, allowing for cross-sex analyses, provided another piece of evidence that the culprit behind the T/T2D association is not T. Despite the PGSs not predicting T levels in the opposite sex, the T2D associations with these PGSs remained similar in both sexes, which means that the T2D associations must stem from variants exerting T-independent effects on these traits (horizontal genetic pleiotropy).

In case of T2D, we also note that by accounting for the SHBG-effects in the MR analysis, the causal estimate of T to T2D dropped substantially, pointing to the potential role of SHBG as a mediator. Similarly, the association of male T PGS to T2D was much weaker after including SHBG in the model. However, in our dataset we detected only a limited amount of direct evidence for SHBG's causality to T2D, echoing some recent findings (54). Collectively, these results thus indicate that although SHBG may partly explain why T associates with T2D, SHBG may not be the only cause for the observed T2D associations.

Additionally:

Supplementary Table 22 now includes an extra sheet that contains full results from multivariable MR in FinnGen, with the effects of total and free T SNPs adjusted for SHBG, and SHBG SNPs effects adjusted for

total and free T. The table shows which causal relationships change after introducing either T or SHBG as a covariate, to better distinguish between the sometimes mixed causality of these traits.

Comment 3

Genetically-determined T accounts for lifelong exposure so it will reflect both organizational and activational T actions. There is an extensive body of experimental data on T administration in humans that precisely characterizes dose-response effects of T within the physiologic range on a variety of relevant, accurately assessed endpoints. Given the limitations of the genetic analyses that the authors now acknowledge, lines 404-419, the manuscript provides limited insight into the contribution of T to disease risk and complex traits.

Author reply:

We, unfortunately, cannot be sure to which extent adult T exposure equals or reflects fetal or childhood T exposure. We need to consider first of all the situation in the womb where the fetus and the placenta are not exposed only to fetal hormones, but also to hormones from the mother, and potential environmental factors affecting these hormones. Secondly, there are extensive changes in the regulation of hormone levels during development, including transient fetal activation of the HPG-axis and reactivation of this axis during the transition from childhood to puberty. These are examples of situations where (due to major changes in physiology) different genetic factors might affect T levels than just those which regulate adult levels. Hence, we have intentionally chosen e.g. to refer to post-pubertal T exposure throughout our manuscript.

Below we discuss these cases in more detail. First, besides the fetus's own genetics, it is established that maternal conditions such as PCOS may affect the hormonal milieu of a developing fetus, also leading to potential health-related issues for the child in the future (DOI: 10.1093/humrep/deaa192). Another example are environmental exposures that may have disproportionately severe effects on fetal hormone levels, including testosterone (<https://doi.org/10.1093/humupd/dmz004>).

Secondly, due to extensive changes in growth patterns upon development, for most traits genetic effects are not fixed during childhood growth. For example in case of BMI and height, the heritability of these traits is dramatically different at different stages of life, and different genetic pathways and genes have been shown to activate and regulate these traits during different stages of childhood (DOI: 10.1038/srep28496, DOI: 10.3945/ajcn.116.130252, DOI: 10.1126/sciadv.aaw3095 and DOI:10.1186/1471-2164-14-547). We expect that the genetics of hormone levels (hormones additionally being subject to extensive modulation throughout life) would likely follow a similar trend. Hence we persist to acknowledge the possibility of adult levels not necessarily accurately reflecting e.g. fetal levels as a potential limitation in our study.

What comes to the second point in this comment, like the reviewer quotes, we readily acknowledge the limitations of our method to study the actions of T in adults in our manuscript, for which the gold-standard method is still RCT's. Nonetheless, we emphasize that our study provides an extensive description on to which extent genetically determined (baseline) differences in adult T levels causally affect various aspects of human health in the general adult population. Comparable studies addressing not only epidemiological but also causal aspects of T function in human health would be infeasible and unethical to conduct on a large scale in a randomised clinical setting. Hence, we believe that genetic data on T levels, allowing for mining of causal relationships, and the wide range of results presented in this manuscript - despite the discussed inherent limitations - are of high value. We now emphasize this in the discussion.

Given the critique on the novelty value of our study by the reviewer and some clever suggestions on how we might improve, we now highlight the novel aspects of our work in more detail in many places in the manuscript. We hope that both the reviewer and the readers are now better equipped to grasp the unique aspects presented in this study. Some of the changes made based on this critique are highlighted in the responses to comment 1, 3 and 4, and a couple of specific changes made based on this comment are listed below.

Changes made to manuscript:

Discussion, page 18, row 337-340:

“Although highly valuable, clinical trials to investigate the relationship of T levels to many complex phenotypes would be often infeasible or unethical to conduct. Here, using genetic data is an alternative means to estimate how T levels affect human health, leading to causal insights beyond the reported epidemiological relationships.”

Discussion, page 23, rows 457:

“Potentially, limitations of our study include that we have concentrated on adult T levels, whereas some of the effects of T might be organizational, acting during specific developmental windows and resulting in permanent phenotypic changes.”

Comment 4

The associations of genetically-determined T with PCOS likely account for its association with hirsutism, infertility and PMB, all features of PCOS.

Author reply: We understand that this might be an intuitive explanation of our findings. We however expect that in our response to the first comment, and with the resulting additions to the manuscript files, we can now show that genetically determined T is different from genetically determined PCOS. Based on our data, PCOS seems not to be the mediator of T’s effects on various phenotypes, and we now hope that the reviewer remains reassured that genetically determined testosterone in women - independently of PCOS - has direct effects on the phenotypes listed above.

Based on the first comment and this comment, we have made the following changes into our manuscript:

Abstract, page 1 rows 22-24: “We find genetically predicted T affects sex-biased and sex-specific traits, with a particularly pronounced impact on female reproductive health across lifespan, including causal contribution to traits like hirsutism and post-menopausal bleeding (PMB), **independent of PCOS**”

Results, page 7, rows 163-166: “With the exception of PCOS, total T and free T displayed similar associations to these traits. **In FinnGen, there was modest case and genetic overlap between PCOS,**

hirsutism and PMB, suggesting the associations of T to latter phenotypes are not PCOS driven (Supplementary Table 21). SHBG adjusted analyses further suggested these associations were strictly androgen dependent (Supplementary Table 21, Supplementary Table 22 and Supplementary Figure 6). “

Results page 10, rows 205-208: Under both MR Egger models, we observed causality between total and free T and PMB and hirsutism ($\beta=0.61$, $p=4.5e-05$ and $\beta=2.11$, $p=0.002$, respectively, for free T in multivariable MR Egger). These effects did not change upon adjusting for the effects of T SNPs on PCOS, further suggesting these phenotypes are primarily androgen driven, and not just simply a consequence of PCOS (Supplementary Table 22). At the same time, MR Egger rather supported causality of total T ($\beta=0.90$, $p=0.012$) and LCV of free T to PCOS (GCP=0.54, $p=0.0017$).“

Discussion, page 18, rows 350-352: Based on our analyses, three major themes emerged regarding T's contribution to disease. First, we report that the studied PGSs associated with disease risk especially in females, including many sex-specific endpoints like hirsutism, PMB and infertility. Importantly, we also show that these effects do not depend on the known link between T levels and PCOS.

Additionally:

Supplementary Table 21 now includes a sheet with a table showing case overlap and genetic correlations between PCOS and related traits in FinnGen R5.

Supplementary Table 22 now includes a sheet with tables containing results on multivariable MR analyses. Here we have used information from the SNPs effects on PCOS and obesity risk to test which causal effects remain for female total and free T, after adjusting for these two conditions.

Reviewers' comments:

Reviewer #4 (Remarks to the Author):

The revised manuscript is improved but its most novel contributions are still not emphasized. It provides a comprehensive evaluation of the contribution of sex-specific genetically-determined T to a variety of complex traits. The findings related to disorders with important sex differences, e.g., cardiovascular and psychiatric conditions, are of considerable interest. Since the findings are mainly negative, a discussion of power should be included in the manuscript. The findings related to female reproductive disorders are almost a distraction. Previous studies have implicated genetically-determined T in the development of PCOS (McAllister et al., 2014, PMID 24706793; Ruth et al., 2020, PMID 32042192). PCOS is the leading cause of elevated T, irregular menses and anovulatory infertility. Increased intracellular T is a prerequisite for the development of hirsutism in the majority of cases. Genetically-determined BMI and fat distribution are causally related to PCOS, female infertility and heavy menstrual bleeding (which likely overlaps with FinnGen PMB) in the UK Biobank (Venkatesh et al., 2022, PMID 35104295) so these traits could confound the reproductive outcomes in the present study. Even if T independently affects the reproductive outcomes, these results would not substantially advance the field.

1. Lack of support for effects independent of PCOS. PCOS is a highly prevalent disorder characterized by increased circulating T levels and chronic anovulation. Because of these endocrine abnormalities, PCOS is the most common cause of hirsutism, irregular menses and/or anovulatory infertility (Azziz et al., 2006, PMID 16940456; Martin et al., 2018, PMID 29522147; Teede et al., 2018, PMID 30052961); a finding confirmed in Finnish women (Taponen et al., 2003, PMID). The diagnostic criteria for PCOS are controversial (Hoeger et al., 2021, PMID 33211867) and variably applied by healthcare providers (Conway et al., 2014, PMID 25049203; Martin et al., 2018, PMID 29522147; Piltonen et al., 2019, PMID 31877155). The coding of the diagnosis of PCOS in electronic health records (EHRs) is inconsistent (Actkins et al., 2021, PMID 32961557; Castro et al., 2015, PMID 26510685). Therefore, it is not possible to exclude PCOS in women with the diagnosis of hirsutism, irregular menses, PMB or female infertility without manual chart review. Even with such review, it may not be possible to determine a diagnosis of PCOS because of lack of relevant data (Castro et al., 2015, PMID 26510685). The absence of PCOS would also be difficult to establish in women with PMB since PCOS is diagnosed in reproductive-age women.

The prevalence of hirsutism and PCOS in the FinnGen population analyzed is well below (both less than 1% based on supplementary table 21) their expected population prevalence rates (Ding et al., 2017, PMID 29221211; March et al., 2010, PMID 19910321; Martin et al., 2018, PMID 29522147) and prevalence rates in other EHR-linked biorepositories (Actkins et al., 2021, PMID 32961557) raising concerns about the accuracy of FinnGen diagnostic coding for these disorders. The current study includes fewer PCOS cases (n=642) than in a recent FinnGen PCOS GWAS (n=797) (Tyrmilä et al., 2021, PMID 34791234). The authors should account for these differences. The analyses of PCOS and hirsutism lacked power because of the small sample sizes of women with these diagnoses. This lack of power combined with the limitations of diagnostic coding constrained the genetic analyses of female reproductive disorders. Accordingly, it is not possible to support the authors' conclusion that genetically-determined T causes hirsutism, female infertility or PMB, independent of PCOS. Statements to this effect should be removed from the manuscript. The potential confounding effects of body weight and fat distribution on female reproductive outcomes should be further discussed in the manuscript (Venkatesh et al., 2022, PMID 35104295).

2. T, SHBG, free T and FAI. The inclusion of both free T and FAI analyses makes the presentation of the results confusing. Both parameters are indices of T that is biologically available to enter tissues. It is well-accepted that non-SHBG T is biologically active (Manni et al., 1985, PMID 4040924;Pardridge, 1986, PMID 3521955;Vermeulen, 2004, PMID 15799123). Free T and FAI are highly intercorrelated as they are both derived from T and SHBG. Free T is the preferred measurement (Rosner et al., PMID 17090633;Vermeulen et al., 1999, PMID 10523012). Reporting results for T, SHBG and free T would clarify the manuscript. The rationale for reporting free T and its intercorrelation with T and SHBG should be outlined. Referring to SHBG as a confounder is also confusing since it is a physiologic regulator of T action and clearly has independent effects on metabolic and reproductive outcomes. Genetically-determined SHBG has protective effects on PCOS risk (Day et al., 2015, PMID 26416764;Ruth et al., 2020, PMID 32042192), which should also be considered.

Response to reviewer's comments

This document provides detailed answers to all the questions raised by the reviewer. We thank the reviewer once more for having the time to go through our manuscript. We have addressed all the comments also in the third review round, and made substantial changes to the manuscript according to the suggestions.

The original reviewer comments are presented *in italics with grey font*, our replies in black, normal type font, and **the changes made to the manuscript with red font**.

Reviewer #4:

Reviewer #4 (Remarks to the Author):

The revised manuscript is improved but its most novel contributions are still not emphasized. It provides a comprehensive evaluation of the contribution of sex-specific genetically-determined T to a variety of complex traits. The findings related to disorders with important sex differences, e.g., cardiovascular and psychiatric conditions, are of considerable interest. Since the findings are mainly negative, a discussion of power should be included in the manuscript. The findings related to female reproductive disorders are almost a distraction. Previous studies have implicated genetically-determined T in the development of PCOS (McAllister et al., 2014, PMID 24706793; Ruth et al., 2020, PMID 32042192). PCOS is the leading cause of elevated T, irregular menses and anovulatory infertility. Increased intracellular T is a prerequisite for the development hirsutism in the majority of cases. Genetically-determined BMI and fat distribution are causally related to PCOS, female infertility and heavy menstrual bleeding (which likely overlaps with FinnGen PMB) in the UK Biobank (Venkatesh et al., 2022, PMID 35104295) so these traits could confound the reproductive outcomes in the present study. Even if T independently affects the reproductive outcomes, these results would not substantially advance the field.

Response to the general comment:

Author reply:

We thank the reviewer from insightful comments also in this round. In response to this general remark, we have made substantial changes into our manuscript. We agree there are several, if not too many, interesting findings in our manuscript given the breadth of the analyses and diversity of diseases studied. We originally chose the female-specific diseases as our primary income angle, since many of these traits have not been thoroughly studied previously with regards to genetically-determined T. We now understand that also the negative findings in our study include a lot of material that might be of interest to many of those working in the field. In the current version of the manuscript, according to the suggestions presented, we have:

- 1) Instead of focusing our abstract, introduction and discussion on the female-specific diseases, we have now put the emphasis of these sections on the metabolic, cardiovascular and psychiatric endpoints in both sexes. Although in our supplementary data and in our manuscript we still present the same results as previously, the manuscript text has been extensively revised, especially from the discussion. We have also condensed the text whenever possible. As an example of these modifications that have taken place during the revision and rewriting, both in the abstract and discussion we now present and discuss the results around the metabolic and cardiovascular disease first, whilst referring, e.g., to female-specific findings later.
- 2) We have modified the title of our manuscript. This now reads as **“Genetic analyses implicate complex links between adult testosterone levels and health and disease”**
- 3) We now mention the possibility of undiagnosed PCOS cases affecting the results, and that we may not be able to fully adjust for such confounding.

Given the substantial changes and complete modification of several sections in the abstract, introduction and discussion, for simplicity, we do not detail these changes here, but these are visible in the track changes version of the manuscript.

1. Lack of support for effects independent of PCOS. PCOS is a highly prevalent disorder

characterized by increased circulating T levels and chronic anovulation. Because of these endocrine abnormalities, PCOS is the most common cause of hirsutism, irregular menses and/or anovulatory infertility (Azziz et al., 2006, PMID 16940456; Martin et al., 2018, PMID 29522147; Teede et al., 2018, PMID 30052961); a finding confirmed in Finnish women (Taponen et al., 2003, PMID). The diagnostic criteria for PCOS are controversial (Hoeger et al., 2021, PMID 33211867) and variably applied by healthcare providers (Conway et al., 2014, PMID 25049203; Martin et al., 2018, PMID 29522147; Piltonen et al., 2019, PMID 31877155). The coding of the diagnosis of PCOS in electronic health records (EHRs) is inconsistent (Actkins et al., 2021, PMID 32961557; Castro et al., 2015, PMID 26510685). Therefore, it is not possible to exclude PCOS in women with the diagnosis of hirsutism, irregular menses, PMB or female infertility without manual chart review. Even with such review, it may not be possible to determine a diagnosis of PCOS because of lack of relevant data (Castro et al., 2015, PMID 26510685). The absence of PCOS would also be difficult to establish in women with PMB since PCOS is diagnosed in reproductive-age women.

The prevalence of hirsutism and PCOS in the FinnGen population analyzed is well below (both less than 1% based on supplementary table 21) their expected population prevalence rates (Ding et al., 2017, PMID 29221211; March et al., 2010, PMID 19910321; Martin et al., 2018, PMID 29522147) and prevalence rates in other EHR-linked biorepositories (Actkins et al., 2021, PMID 32961557) raising concerns about the accuracy of FinnGen diagnostic coding for these disorders. The current study includes fewer PCOS cases (n=642) than in a recent FinnGen PCOS GWAS (n=797) (Tyrimi et al., 2021, PMID 34791234). The authors should account for these differences. The analyses of PCOS and hirsutism lacked power because the small sample sizes of women with these diagnoses. This lack of power combined with the limitations of diagnostic coding constrained the genetic analyses of female reproductive disorders. Accordingly, it is not possible to support the authors' conclusion that genetically-determined T causes hirsutism, female infertility or PMB, independent of PCOS. Statements to this effect should be removed from the manuscript. The potential confounding effects

body weight and fat distribution on female reproductive outcomes should be further discussed in the manuscript (Venkatesh et al., 2022, PMID 35104295).

Author reply:

Spurred by this comment #1, we have consulted a PCOS expert MD PhD Terhi Piltonen, professor in Obstetrics and Gynecology and reproductive endocrinology in University of Oulu, Finland, about the results, and have added her as a co-author into our study. Based on these discussions and the comment, we have now omitted all mentions that the associations to female-specific traits are fully independent of PCOS, and considerably softened our interpretations on the subject. Instead, we now explicitly mention the possibility of undiagnosed PCOS cases (as previously speculated by Dr. Piltonen and colleagues, Tyrmi et al. 2022, DOI: 10.1093/humrep/deab250), and the overall complex nature of the syndrome affecting the results (e.g. Dapas et al. 2020, <https://doi.org/10.1371/journal.pmed.1003132>). We understand that our data is complicated by the likely incomplete nature of the registry data in Finland, and for example by the possibility that the treating clinician may have diagnosed hirsutism, without checking for underlying PCOS and vice versa. We have therefore carefully checked and removed all direct statements that genetically-determined T would be - independent of PCOS - directly responsible for phenotypes such as hirsutism, infertility and PMB. We still report in the supplementary results that genetic data shows these diagnoses do not completely overlap in FinnGen, while we also acknowledge that this may reflect variable diagnosis practices rather than true biological differences between these conditions.

Besides this main issue in this question, we have also addressed the other points introduced by the reviewer in our response below, and modified the manuscript accordingly.

Changes made to the manuscript:

Abstract, page 1, removed a statement “*independent of PCOS*”, instead now state: “*PCOS-related traits like hirsutism, and post-menopausal bleeding (PMB)*”.

Results page 7, rows 156-160 when talking about these findings we now mention: *“Of the novel endpoints, we robustly linked T with hirsutism and post-menopausal bleeding (PMB) (HR=1.45, p=2.7e-08 and HR=1.05, p=0.00032 for free T PGS), especially the former associating with PCOS and the latter with endometrial cancer (31, 32)*

Results page 7, rows 162-163 replaced a sentence: “suggesting the latter phenotypes are not fully PCOS driven” with *“potentially reflecting underdiagnosis of PCOS in the dataset”*, referencing a study by Tyrmi et al. 2022 (DOI: 10.1093/humrep/deab250)

And the results page 10 rows 203-205 now reads as *“Adjusting for PCOS had negligible effects on the MR results, suggesting these phenotypes are primarily dependent on androgen load (Supplementary Table 22).”*

Discussion, page 19 rows 361-368 now reads as:

“Finally, we report that the studied PGSs associated with disease risk especially in women, including PCOS-related endpoints like hirsutism, PMB and infertility. We unfortunately cannot state if genetically determined T has PCOS-independent effects on these traits, since our study may be limited by PCOS being potentially underdiagnosed in FinnGen, or the diagnosed PCOS cases representing only a subpopulation of this heterogenous disorder (47, 48). In addition, all these reproductive traits correlate with obesity (49), that in turn correlates with PCOS and T levels (Supplementary Figure 9 and Supplementary Table 22), and despite our efforts we may have not been able to fully account for such confounding in our study. ” with three added references (Actkins et al., 2021, PMID 32961557, Dapas PMID: 32574161 and Venkatesh et al., 2022, PMID 35104295).

And in the discussion of study limitations, rows 425-428, we acknowledge

A special issue for genetic studies is susceptibility to confounding by pleiotropy, whereby a gene influences multiple traits via independent biological pathways (34). Although in our analyses we opted for adjusting for the effect of SHBG, BMI, PCOS and menopause in females – (Supplementary Files 21-27 and Supplementary Figure 9) (57-60) - such factors may still affect our findings.

With the above changes we aim to have replied also to the following comment:

The potential confounding effects body weight and fat distribution on female reproductive outcomes should be further discussed in the manuscript (Venkatesh et al., 2022, PMID 35104295)

Responses to the two final points in this comment #1:

The current study includes fewer PCOS cases (n=642) than in a recent FinnGen PCOS GWAS (n=797) (Tyrmi et al., 2021, PMID 34791234). The authors should account for these differences

What comes to the discrepancy between sample counts in Tyrmi et. al and our study, there is a simple explanation. Whereas we used FinnGen release 5, i.e., the same data release as the upcoming FinnGen flagship paper, the Tyrmi study used release 6 with includes over 40,000 more samples and therefore more PCOS cases.

The analyses of PCOS and hirsutism lacked power because the small sample sizes of women with these diagnoses.

In supplementary Figure 5, we show that we have power to detect large effects even for these endpoints (HR>1.3), but like mentioned above we now readily acknowledge that potential underdiagnosis of PCOS in our cohort may affect the results. We now clearly mention what size of effects for these rare endpoints we had power to detected in several points in the manuscript, including:

Results page 6, rows 135-137

“Power analyses supported our ability to detect large effects (HR> 1.3) even for the rarest, and relatively small effects (HR>1.05) on more common phenotypes (Supplementary Figure 5 and Methods).”

Discussion, rows 352-355

“Despite the rarity of some of the studied phenotypes, we estimate that we were well powered to reliably detect large effects on all endpoints (>30% increase in relative risk), although we may have missed any subtler effects (<5 % increase in relative risk) in case of the rarest phenotypes (Supplementary Figure 5).”

2. T, SHBG, free T and FAI. The inclusion of both free T and FAI analyses makes the presentation of the results confusing. Both parameters are indices of T that is biologically available to enter tissues. It is well-accepted that non-SHBG T is biologically active (Manni et al., 1985, PMID 4040924;Pardridge, 1986, PMID 3521955;Vermeulen, 2004, PMID 15799123). Free T and FAI are highly intercorrelated as they are both derived from T and SHBG. Free T is the preferred measurement (Rosner et al., PMID 17090633;Vermeulen et al., 1999, PMID 10523012). Reporting results for T, SHBG and free T would clarify the manuscript. The rationale for reporting free T and its intercorrelation with T and SHBG should be outlined. Referring to SHBG as a confounder is also confusing since it is a physiologic regulator of T action and clearly has independent effects on metabolic and reproductive outcomes. Genetically-determined SHBG has protective effects on PCOS risk (Day et al., 2015, PMID 26416764;Ruth et al., 2020, PMID 32042192), which should also be considered.

Author reply:

We appreciate these points raised by the reviewer. Although we initially thought that some potential readers might like to see data from both free T and FAI to compare our results to previous data, we do agree that concentrating only to one of these would make the manuscript easier to follow. **Accordingly, we have now removed discussion on FAI in the introduction and results sections and from the main figures, and concentrated only on reporting results for free T.** We have included results for FAI only as supplementary data. In the current version, we also emphasize the relationship of T, SHBG and free T in several instances in the manuscript, and clearly mention the close relationship between free T and FAI. Finally, based on this comment, we have also changed the text in several places to refrain from referring to SHBG merely as a confounder of T action, and currently point towards its potential direct role on some phenotypes, like suggested.

Changes made to the manuscript:

We now report the rationale for reporting the results for free T in the introduction with an added reference:

Introduction, page 2, rows 48-50: "Free T is considered to represent the most potent form of T in terms of biological activity, and although extensively debated, may have different clinical significance than total T"

Like the reviewer states and suggests, we bring forward the close relationship between free T and FAI, and have removed all discussion on FAI from the main text.

Methods, page 24, rows 477-479: "For clarity, given the close relationship between free T and FAI, in the results section we concentrate on describing the results for free T, whereas the results for FAI are available in the supplementary data".

In summary, in response to the main issue raised in the comment 2, we have thus now removed all mentions of results on FAI, and report only the results for free T in the main text. This includes updates on figures 2 and 5.

Specific changes:

We have removed all mentions to FAI and results associated with FAI, and replaced these with free T where applicable, making changes to several of the main figures. These changes are listed below, and are visible in the track changes version of the manuscript:

Introduction: row 59

Results: rows 80, 108, 115, 117, 165, 279, 285

Complete revisions of Figure 1, Figure 2 and Figure 5 and figure legends.

Besides excluding FAI from the results, the other key point raised by the reviewer in this comment was to refrain from referring to SHBG as a "confounder". We have now removed all instances where we mention that SHBG may "confound" T's associations and instead now clarify that SHBG may have direct, independent effects on phenotypes. For example, instead of talking about confounding, we now write in results (from page 7, rows 165-167):

“In contrast, many associations detected for female free T like infertility risk (HR=1.04, p=0.00502), seemed highly dependent on SHBG (SHBG showing generally favourable effects on reproductive traits, e.g., HR=0.98, p=0.00116 for irregular menstruation).”

Additionally, e.g. row 196: confounded by SHBG -> *attributable to SHBG*

And in the discussion, instead of mentioning confounding of T’s effects, we now refer to direct effects of SHBG on many phenotypes:

Page 18, Rows 355-357 now reads as: *“Since total T in men correlates with SHBG levels, this underscores potential contribution of SHBG to these associations (13-15). Especially for many metabolic traits SHBG – either directly or through its metabolic network - appeared to modify some of T’s associations and causality estimates.”*

To prevent misunderstandings, we have omitted referring to confounding in row 427, and modified rows 588-589 in methods.

We hope that with the above changes, we have addressed all the issues that were raised during the previous round of revision.

Reviewers' comments:

Reviewer #5 (Remarks to the Author):

The authors have revised their work, taking into account the comments received from initial referee-4. While actions taken have helped to expand the scope and potential impact of the study, this referee believes that the study still falls somehow short in providing conclusive evidence for some of the major conclusions, due to potential flaws in interpretation of data (in the appropriate biological context) and/or technical limitations.

1. From a conceptual standpoint, I believe it is crucial to more explicitly discriminate between genetically-determined T or SHBG levels, as addressed in the study, and actual influence of T exposure upon the different traits under analysis. While basal T levels might be defined to some extent by a certain genetic make-up, it is also known that T fluctuate quite substantially in a rather dynamic manner, under different physiological and stress conditions, such as fasting or long-term obesity. These non-genetic factors (e.g., environmental, nutritional) can profoundly alter the actual acute and long-term exposure to T and this phenomenon, which is not address by the current study, may render some conclusions of limited value. This issue is relevant and should be more clearly emphasized in the MS, since at some places, it seems to this referee that the authors equal genetic determinants of T levels (as PGS) to actual T exposures, and this direct association might be greatly perturbed by numerous confounding factors, not accounted for in this study.

2. Genetic associations are based on single T and SHBG determinations, in I am not mistaken, that would not capture the dynamic fluctuations indicated above and this may limit the overall impact and strength of some conclusions, including the causative associations (or their lack of). In addition, T determinations are based on immunometric assays, which might be of limited accuracy in the case of females. Could this be the cause for underdiagnosis of PCOS/hyper-androgenemia in women in the Finnish cohort? As additional concern, the fact that some key hormonal determinations were conducted between 2007-2011, when immunometric assays might not be as accurate as current MS-based methods, brings additional concerns on the validity of these hormonal determinations of T levels in females.

3. The authors define potential genetic determinants of free T levels, which is actually a construct defined by total T and SHBG levels, but whose biological meaning is still the subject of debate. In fact, since free T is defined by the other two variables, this referee wonders whether independent discussion of genetic factors for the three variables is appropriate or not (as they are connected to each other; changes in T and/or SHBG influence free T).

4. Some of the discussions of the data do not sufficiently take into account previous knowledge on the distinct (if not opposite) interplay between T levels and metabolic disease in males vs. females. In fact, it is well known that lowering of T levels increases the risk of metabolic disease in men, while in women, the opposite hold true (increasing androgen levels raises the risk of metabolic impairment, as in PCOS). Hence, the cross-sex analysis of the impact of genetically-defined T levels (or PGS) on specific (metabolic) traits should take this into consideration, as in fact, this opposite behavior is predictable on the basis of previous literature.

Reviewer #6 (Remarks to the Author):

The manuscript is a sophisticated mathematical analysis of UK Biobank and a Finnish study with little real-world reference for validation. The fatal methodological flaws in this manuscript undermine its credibility for anyone with expertise in reproductive endocrinology.

1. Both studies measured serum testosterone by immunoassays. Modern clinical research on reproductive steroids research relies solely on reference liquid chromatography mass spectrometry methods. It is already widely and well known that the old (but cheap) immunoassay technology is passably accurate for the higher levels prevailing in men but they are little more than random number generation at the low level in women at any age and in children. See for example a famous editorial from 2 decades ago Herold DA, Fitzgerald RL 2003 Immunoassays for testosterone in women: better than a guess? Clin Chem 49(8):1250-1251. The dichotomy on sex that these analyses purport to show could be simply due to the invalidity in women but not men.

2. To compound the measurement methodology problems, the authors use so-called "free" testosterone (or sometimes confusingly refer to "bioavailable" testosterone which is a different entity) as if it is a real analyte. In fact, it is simply an inaccurate calculation purporting to replicate lab measurements but quite systematically inaccurate in doing so. It has formulaic dependence on testosterone and SHBG making it unsuitable collinear to include in any models with its component variables. Moreover, for women serum T varies little, but serum SHBG varies widely with age and other influences. Hence for women the "free" testosterone is simply a masked reciprocal of serum SHBG.

3. The authors fail to validate the many predictions of these models for sex-dependent effects with objective independent real world findings on androgen dependent variables. It would take me too much time for this short overview to outline these, but in their absence one learns not to rely on such over-engineered abstract modelling.

Reviewer #7 (Remarks to the Author):

The authors have answered all prior critiques, which has significantly improved the manuscript.

Author rebuttal

We thank all the reviewers for taking the time to go through our manuscript and the comments. The major criticism from reviewers #5 and #6 concerns the validity of the testosterone (T) measures in the UK Biobank data. We acknowledge there are more accurate methods for measuring blood testosterone levels than the immunometric assay applied in the UK Biobank sample. This method is more cost-effective and a faster option and thus better suited for testosterone profiling in large samples, yet, like the reviewers note, such assays may be more inaccurate in the lower range of serum T values. However, the concern raised by the reviewers, reviewer #6 in particular, that the testosterone measurements in women provide nothing but random numbers and therefore are of no use, is fully unfounded. This is supported by several observations:

- We find more than 100 genomic regions highly significantly ($p < 5e-8$) associated with testosterone levels in women in the UK Biobank population. Should the T measurements be merely from a random number generator, finding such a number of - or any – associations would be extremely unlikely given the stringent p-value threshold applied. These genetic associations provide solid evidence that the testosterone measurements from the UK Biobank are legitimate and biologically informative in both sexes.
- The identified genomic regions, for both female and male T levels, implicate multiple genes with established functions in steroid biology, such as *CYP3A7*, *CYP17A1* and *LHB* identified in the UK Biobank females. In other words, the validity of the genetic findings based on immunohistochemical measurements is also supported by prior experimental research.
- Providing further evidence for the value of the UK Biobank testosterone data, in our genetic analyses we replicate previously discovered genetic loci for testosterone. We also find highly concordant effect sizes with genome-wide association analyses from independent cohorts where a considerable fraction of samples have testosterone measurements coming from methods other than immunometric assays (i.e., mass spectrometry, preferred by the reviewers) (e.g., Ohlsson et al. 2011 (<https://doi.org/10.1371/journal.pgen.1002313>), Pott et al. 2019 (<https://doi.org/10.1210/jc.2019-00757>)). Additionally, our genetic findings, including the testosterone polygenic scores built using the genetic associations from the UK Biobank analyses, replicate in the Young Finns cohort, as displayed in Supplementary Table 20 in the manuscript.
- While we agree that the immunoassays in females may be unsuitable for diagnostic purposes given their challenges (lower accuracy, and potential bias to either direction in average values compared to mass spec assays), these biases do not represent a major concern in our study. We focus on population-level variability i.e. order of samples, rather than absolute values at individual level. Earlier comparisons have shown high correlations between the two technologies, meaning that both technologies reliably reflect this order or the samples. For instance the paper referred in Herold DA, Fitzgerald RL 2003, (<https://doi.org/10.1373/49.8.138>) shows that the average correlation with immunoassays and the reference assay for female testosterone was as high as $R=0.76$, with the best immunoassay showing a correlation of 0.89 in females.
- We now present new analyses to further support the validity of the UK Biobank testosterone data, and to demonstrate daily or environmental fluctuation is not a major determinant or confounder in these testosterone measurements. To this end, we have assessed the correlation of testosterone levels between two timepoints in the UK Biobank population, from a baseline and a follow-up visit for 7,097 males and 5,285 females, with the original samples collected 2006-2010, and additional samples taken in 2012-2013, T measurements being performed from these blood samples from 2015 onwards. Reassuringly, in these data, in both males and females, the testosterone measurements from roughly five years apart are highly correlated ($R=0.678$ in males, $R=0.709$ in females). These high correlations indicate that the immunoassay method used reliably captures population-level variability in testosterone

levels and further show that there exists a fairly stable individual-level baseline for testosterone, likely largely heritable in basis, reflecting an individual's life-long T exposure. In fact, testosterone measured during the second visit predicts the testosterone level at the first visit far better than for instance BMI at the first visit ($R=-0.295$ in males, $R=0.081$ in females), which we additionally already take into account as a confounder in our analyses.

To more clearly acknowledge that our data is based on immunoassays, and to better convey their potential limitations, we now bring this forward in our manuscript in several places in the results and discussion section. We illustrate the method and its potential limitations in the results (rows 79 and 106->), discussion (418-> and 427->) and methods (row 474->). At the same time we also discuss the evidence that this is not a major concern for our study, and show and discuss the high correlation between testosterone measurements between to time points in the UK Biobank, likely reflecting the heritable baseline in T levels.

To better communicate the value and the implications of the genetic discoveries we have made changes to the manuscript. In several places we now emphasise that our results reflect especially the consequences of genetically determined T levels, starting from abstract (rows 18 and 20), having made changes also especially to discussion (rows 341, 344, 349, 352, 358, 391, 393, 401, 405, 431, and 448).

We also clarify the value of the genetic data from Row 106 -> *Reflecting the high heritability, T measurements from two different time points were highly correlated in both sexes (Supplementary Figure 3). Collectively, despite being based on immunoassays which may have limited use in clinical settings especially in females, these findings illustrate that the UK Biobank data permits construction of robust genetic instruments to study how post-pubertal T and SHBG levels relate to adult health.*

As the major criticism in general seems to concern the choice of methods and validity of the data, we are also slightly worried that we have not been able to communicate the value of the genetic analyses in the best possible way. This may be partly due to our selection to include a combination of several different analyses into our manuscript.

As a result, we have made changes to better show the meaning and value of the PGS-analyses, including changes in text (results rows 132-133) and modified Supplementary Figure 5. We have also revised the legend of main figure 1 to better convey our study flow (rows 94->).

The key idea in our study (associations of T / SHBG / free T PGS against disease endpoints in FinnGen) is that we use genetic markers associated with testosterone (obtained in the UK Biobank analysis, with the immunoassays) to predict (genetically-determined) serum T in the FinnGen data. I.e., we predict the population-level variability in T in FinnGen but not actual T levels. Having ranked persons according to their genetic predisposition to higher and lower T, we can then assess the relationship between (genetically-determined) adult T and diverse clinical endpoints in FinnGen, suggesting potential causal relationships. In short, these analyses thus answer the question that what are the consequences of having a genetic predisposition to higher or lower T levels. **We now state this in the manuscript results section (row 132->), and explain this rationale in Supplementary Figure 5.**

Then, we extend these analyses using Mendelian Randomization (MR), helping us to obtain further estimates on direct causality of T and SHBG on complex traits. MR, in simple terms, compares the effect sizes of genetic variants associated with an exposure (T in this case) with their effect sizes on outcome (endpoints). These analyses, in combination with the cross-sex analyses, have power to show if the detected PGS associations reflect direct action of T, or if these stem from the genetic variants associated with T and SHBG having other, (pleiotropic) effects e.g. on distinct metabolites or physiological pathways. Additionally, the MR strategies allow for direct estimation of population level consequences of 1SD increase in T levels.

We emphasize that in our manuscript we currently extensively discuss the limitations of these genetic approaches, including that the PGSs explain only a proportion of the variation in adult T levels. We still have more than adequate power to make conclusions about the phenotypic consequences of population level variation in baseline T levels (especially in the normal physiological range), and to speculate what kind of effects and effect sizes we could expect to observe per 1SD higher T / SHBG levels. We hope that we are not unintentionally confusing critical readers by claiming these results would make void results e.g. from studies addressing acute T actions by manipulating the baseline hormonal balance, or from cases where there is a clear biological deficit related to T action.

Reviewer #5 (Remarks to the Author):

The authors have revised their work, taking into account the comments received from initial referee-4. While actions taken have helped to expand the scope and potential impact of the study, this referee believes that the study still falls somehow short in providing conclusive evidence for some of the major conclusions, due to potential flaws in interpretation of data (in the appropriate biological context) and/or technical limitations.

- 1. From a conceptual standpoint, I believe it is crucial to more explicitly discriminate between genetically-determined T or SHBG levels, as addressed in the study, and actual influence of T exposure upon the different traits under analysis. While basal T levels might be defined to some extent by a certain genetic make-up, it is also known that T fluctuate quite substantially in a rather dynamic manner, under different physiological and stress conditions, such as fasting or long-term obesity. These non-genetic factors (e.g., environmental, nutritional) can profoundly alter the actual acute and long-term exposure to T and this phenomenon, which is not address by the current study, may render some conclusions of limited value. This issue is relevant and should be more clearly emphasized in the MS, since at some places, it seems to this referee that the authors equal genetic determinants of T levels (as PGS) to actual T exposures, and this direct association might be greatly perturbed by numerous confounding factors, not accounted for in this study.*

We agree and acknowledge these issues as a limitation in our manuscript. **We therefore extensively discuss over several paragraphs about the strengths and the limitations of the genetic methods to draw robust conclusions about T and SHBG action in our manuscript (rows 418-453).**

In response to this comment, we have added data to show that despite the numerous factors that can affect T levels and T secretion, the baseline T levels remain in fact relatively stable over the course of time in the adult population, reflecting heritability of T. **We now show that there exists a significant and robust correlation between an individual's adult T levels measured in two different occasions (Supplementary Figure 3 and Methods rows 483 ->).** The data includes measures from initial and replication visits on average 5 years later for a subset of UK Biobank participants (N males = 7097 and N females = 5285, please see Figure 1). For males, we observed a correlation of $R = 0.678$ and for females $R = 0.709$ between the T measures from two different time points. Interestingly, this is considerably higher than for example correlation of T levels and BMI at a single time point (for males, $R = -0.295$, and for females $R = 0.081$). We would also like to stress that although several non-genetic factors can definitely alter even the long-term T exposure over the course of life, our methods allow for making conclusions about the potential consequences of any long-term alterations in T levels in the normal physiological range.

Figure 1. Correlations between T measurements from 2006-2010 and 2012-13 visits for a subset of UK Biobank participants

Additionally, this comment spurred us to pay special attention to precisely mentioning that we indeed have focused on studying the consequences of genetically determined, baseline T levels. We thus have not studied acute T action, or the consequences of altering this baseline T / SHBG levels by interventions. **We now emphasize this right from the first paragraph of the abstract (rows 18 and 20), having made changes also especially to discussion (rows 341, 344, 349, 352, 358, 391, 393, 401, 405, 431, and 448).**

We hope we do not confuse the readers with our choice of words here anymore.

2. *Genetic associations are based on single T and SHBG determinations, in I am not mistaken, that would not capture the dynamic fluctuations indicated above and this may limit the overall impact and strength of some conclusions, including the causative associations (or their lack of). In addition, T determinations are based on immunometric assays, which might be of limited accuracy in the case of females. Could this be the cause for underdiagnosis of PCOS/hyper-androgenemia in women in the Finnish cohort? As additional concern, the fact that some key hormonal determinations were conducted between 2007-2011, when immunometric assays might not be as accurate as current MS-based methods, brings additional concerns on the validity of these hormonal determinations of T levels in females.*

The reviewer is correct that the genetic associations are based on single determinations of T and SHBG, and that the genetic findings identified therefore do not reflect any dynamic fluctuations, for example, the potential effects of acute changes in the secretion of sensitivity to testosterone. Related to our response to the first comment, we have therefore made the best effort to highlight in the manuscript that we are in fact studying the long-term consequences of average genetically determined T levels. **E.g., we use the words “post-pubertal T exposure” to distinguish our testosterone variable from short-term changes in T levels. We acknowledge this as one possible limitation of our study and accordingly discuss this in the manuscript (e.g., rows 423-> and 444->).** However, we remind that even with this T variable we nevertheless find and replicate several previously established links between T and different phenotypes, ranging from baldness and hirsutism to hormonal cancers and hemoglobin levels, showing genetic data indeed is a valid instrument to assess the long-term consequences of T levels.

What comes to the limitations of the immunometric method, we are aware that this is not the most accurate method of measuring T levels and can provide biased estimates for serum T particularly in

females. While the immunometric T measurements may be suboptimal for diagnostic purposes, we argue the potential bias in T concentration is of lower relevance for genetic analyses like ours. For example in the PGS analyses we do not use the absolute T values, but rather use the rank of samples within each sex, i.e. the T data is inverse rank normalized prior to genetic analyses as the variable of interest. Accordingly, we can clearly show that when applied to large cohorts like the UK Biobank, this immunometric method can definitely produce valuable, biologically relevant data (please refer to the general comments in the beginning of the rebuttal letter for details). We hope that also with the new data that we present regarding the correlation between T measurements several years apart also serves to reassure the reviewer that the immunometric method is indeed a sufficiently valid method for these genetic analyses.

In response to this criticism we clearly acknowledge that our data is based on immunoassays, and to better convey their potential limitations, we now bring this forward in our manuscript in several places in the results and discussion section. We illustrate the method and its potential limitations in the results (rows 79 and 106->), discussion (418-> and 427->) and methods (row 474->). At the same time we also discuss the evidence that this is not a major concern for our study.

We would also like to point out that the choice of the technique for T measurements in the UK Biobank is unconnected to the underdiagnosis of PCOS in FinnGen. The key idea in our analyses is that we only use genetic markers associated with testosterone (obtained in the UK Biobank analysis, with the immunoassays) to predict (genetically-determined) serum T in the FinnGen data, i.e., we predict the population-level variability in T in FinnGen but not actual T levels. This linkage via genetic markers then allows us to assess the relationship between (genetically-determined) T and diverse clinical endpoints in FinnGen, including PCOS. The PCOS diagnoses are determined by treating clinicians from different Finnish hospitals with the appropriate Finnish diagnostic standards, and at least in the present day serum T in women is preferably measured via mass spectrometry.

3. *The authors define potential genetic determinants of free T levels, which is actually a construct defined by total T and SHBG levels, but whose biological meaning is still the subject of debate.*

In fact, since free T is defined by the other two variables, this referee wonders whether independent discussion of genetic factors for the three variables is appropriate or not (as they are connected to each other; changes in T and/or SHBG influence free T).

We openly discuss the relationship between T, SHBG and calculated free T measurements in the manuscript, and how we have taken this into account in our **study (e.g., rows 47->, 117->, 149-> 365-> and 434->)**. We have additionally performed many analyses to tease out the relationship between T, free T and SHBG, including adjusting total and free T's associations and MR estimates with effects on SHBG (**Row 149->, and Supplementary table 21**). **We state the controversy surrounding free T already in the introduction (row 48-> "Free T is considered to represent the most potent form of T in terms of biological activity, and although extensively debated may have different clinical significance than total T"). In response to this comment we, for example, now additionally state in rows 170-171 "In contrast, reflecting the close relationship between SHBG and calculated free T in females, some associations detected for female free T like infertility risk (HR=1.04, p=0.00502), seemed highly dependent on SHBG"**.

Since the relationship between T, SHBG and free T is an important topic like the reviewer suggests, (and mentioned also in the comment 2 by reviewer #6), we have still clarified the discussion over the subject. Rows 365-373 now reads as: "Secondly, we highlight how our data reflects the known correlations between total T, SHBG and free T levels in both sexes. Total T in men correlates with SHBG levels (13-15), and the traits are also highly correlated at the genetic level. Underscoring the potential contribution of SHBG to total T associations, we found SHBG

causal for total T levels in males, and not vice versa. In males, especially for many metabolic traits, SHBG – either directly or through its metabolic network – indeed appeared to modify some of total T’s associations and causality estimates. Consistent with potentially divergent biological effects for the SHBG bound and unbound T, we however observed distinct association profiles for total and free T fractions in males. In females, the situation was the opposite: whereas total T associations seemed largely independent of SHBG action, for many metabolic traits the high dependency of calculated free T on SHBG levels was evident.”

Although we do report for instance genetic correlations (reflecting whether the genetic variants affecting SHBG, T and free T are shared and act similarly), we have currently chosen not to explain in detail how the genetic variants for these three traits differ, as the focus of our manuscript is in the disease associations and estimating causality. We appreciate that also the usefulness and validity of calculated free T levels continues to be a debate in itself, and this is definitely not meant to be the focus of our manuscript. We however currently see it as relevant to include calculated free T as one of the studied factors in our paper, since several other papers (including Ruth et al. 2020 and Mohammadi et al. 2021) have recently done so.

4. *Some of the discussions of the data do not sufficiently take into account previous knowledge on the distinct (if not opposite) interplay between T levels and metabolic disease in males vs. females. In fact, it is well known that lowering of T levels increases the risk of metabolic disease in men, while in women, the opposite hold true (increasing androgen levels raises the risk of metabolic impairment, as in PCOS). Hence, the cross-sex analysis of the impact of genetically-defined T levels (or PGS) on specific (metabolic) traits should take this into consideration, as in fact, this opposite behavior is predictable on the basis of previous literature.*

In response to this comment we now try to emphasize the known interplay between T and metabolic disease in males and females even better in our manuscript. **We now clearly state this link in the manuscript (Discussion row 354->): “The results generally echo the epidemiological observations that lower T levels correlate with increased risk of metabolic disease in men, while in women the opposite hold true.”**

We however want to emphasize that the purpose of our cross-sex analyses was not to test for the known, opposite associations between T and metabolic health in males and females. Instead, these were designed to test if the PGS associations truly reflect T action. The key idea here was that – uniquely amongst the complex traits - the genetic variants that raise T levels in men do not affect T levels in women and vice versa. Therefore, any associations for these sex-specific PGSs that replicate in the opposite sex cannot be caused directly by T. Rather, these likely reflect underlying genetic pleiotropy (i.e some variants associated with T have other effects on metabolism instead of being restricted to primarily affecting T levels).

Reviewer #6 (Remarks to the Author):

The manuscript is a sophisticated mathematical analysis of UK Biobank and a Finnish study with little real-world reference for validation. The fatal methodological flaws in this manuscript undermine its credibility for anyone with expertise in reproductive endocrinology.

1. *Both studies measured serum testosterone by immunoassays. Modern clinical research on reproductive steroids research relies solely on reference liquid chromatography mass spectrometry methods. It is already widely and well known that the old (but cheap) immunoassay technology is passably accurate for the higher levels prevailing in men but they are little more than random number generation at the low level in women at any age and in children. See for example a famous*

editorial from 2 decades ago Herold DA, Fitzgerald RL 2003 Immunoassays for testosterone in women: better than a guess? Clin Chem 49(8):1250-1251. The dichotomy on sex that these analyses purport to show could be simply due to the invalidity in women but not men.

Our response to this criticism is available in the beginning of the rebuttal letter as a general comment. We hope this helps to convince this critical reviewer that especially when applied over a large number of samples, immunoassays can provide valid, biologically relevant data reflecting variability in T levels in a population, and therefore that the use of these assays in the UK Biobank does not constitute a fatal methodological flaw.

2. *To compound the measurement methodology problems, the authors use so-called “free” testosterone (or sometimes confusingly refer to “bioavailable” testosterone which is a different entity) as if it is a real analyte. In fact, it is simply an inaccurate calculation purporting to replicate lab measurements but quite systematically inaccurate in doing so. It has formulaic dependence on testosterone and SHBG making it unsuitable to include in any models with its component variables. Moreover, for women serum T varies little, but serum SHBG varies widely with age and other influences. Hence for women the “free” testosterone is simply a masked reciprocal of serum SHBG.*

We are aware and openly discuss the measurements, their relationships and their problems throughout the manuscript (**rows 48->, 106-> 365-> and 434->**). Besides the T/SHBG/free T relationship in females, interpreting the role of SHBG behind male total T associations seemed to be critical, these two traits sharing many genetic factors.

We aim to open up this relationship in many parts of our manuscript, having used several different analyses to tease out the relationship between these traits. For example, concerning this relationship in females: **rows 170-171 “In contrast, reflecting the close relationship between SHBG and calculated free T in females, some associations detected for female free T like infertility risk (HR=1.04, p=0.00502), seemed highly dependent on SHBG”.**

and males e.g. row 198-> **“Finally, we distinguished between the effects of T and SHBG by including the latter as a covariate in multivariable MR Egger analyses (37). Despite the many associations (Figure 2), we observed no clear evidence for T being directly causal for metabolic traits.”** Please refer also to the response to reviewer #5 comment 3.

In response to this comment and comment 3 by reviewer #5, we have also clarified the discussion over the subject (rows 365-373 now reads as: “Secondly, we highlight how our data reflects the known correlations between total T, SHBG and free T levels in both sexes. Total T in men correlates with SHBG levels (13-15), and the traits are also highly correlated at the genetic level. Underscoring the potential contribution of SHBG to total T associations, we found SHBG causal for total T levels in males, and not vice versa. In males, especially for many metabolic traits, SHBG – either directly or through its metabolic network – indeed appeared to modify some of total T’s associations and causality estimates. Consistent with potentially divergent biological effects for the SHBG bound and unbound T, we however observed distinct association profiles for total and free T fractions in males. In females, the situation was the opposite: whereas total T associations seemed largely independent of SHBG action, for many metabolic traits the high dependency of calculated free T on SHBG levels was evident.”

3. *The authors fail to validate the many predictions of these models for sex-dependent effects with objective independent real world finding on androgen dependent variables. It would take me too much time for this short overview to outline these, but in their absence one learns not rely on such over-engineered abstract modelling.*

We are sorry the reviewer does not specify these instances where our data does not validate previous results, which would better allow us to address this criticism. In the absence of concrete examples to discuss, we respond to this comment in more general terms.

First, while the reviewer calls our approach as abstract modelling, we argue that our data and findings are, in fact, highly relevant to “real world” settings. One purpose of our study is actually to validate the many epidemiological findings that would be impossible or unethical to validate via randomised clinical trials. Hence large-scale studies such as ours, extending the epidemiological data by addressing how heritable differences in T / SHBG associate and causally relate to phenotypes at the population level, are of high value. We also would like to point out that - consistent with our findings and considering the 7-15x differences in T levels between an average male and female (that should lead to vast differences in disease incidence between the sexes if adult T levels were the major or sole culprit for diseases and diagnoses) - it is unlikely that varying adult T levels within sex have huge effects on most complex traits. And this is exactly what we detect. **We now also state this in the discussion (rows 402-404).** Compellingly, although not surprisingly, we instead link heritable differences in T with causal effects on traits and phenotypes with clear sex differences or known sexual dimorphism.

Reviewer #7 (Remarks to the Author):

The authors have answered all prior critiques, which has significantly improved the manuscript.

We greatly thank the reviewer from his/her time and comment.

REVIEWERS' COMMENTS:

Reviewer #5 (Remarks to the Author):

The authors have thoroughly considered the major comments and criticisms of this referee. While the technical limitation of the immunometric assays persists, I do believe that the careful consideration and discussion of such limitation, along with the revisions undertaken to address the other points included in my previous evaluation, have significantly improved the MS, making clearer the focus and actual findings of the current paper.

Author response

REVIEWERS' COMMENTS:

Reviewer #5 (Remarks to the Author):

The authors have thoroughly considered the major comments and criticisms of this referee. While the technical limitation of the immunometric assays persists, I do believe that the careful consideration and discussion of such limitation, along with the revisions undertaken to address the other points included in my previous evaluation, have significantly improved the MS, making clearer the focus and actual findings of the current paper.

Author Reply:

We owe many thanks for the reviewer for taking the time to help us improve our manuscript.